# Activation of NIX-mediated mitophagy by an interferon regulatory factor homologue of human herpesvirus

Mai Tram Vo [1], Barbara J. Smith[2], John Nicholas[1] & Young Bong Choi [1]

Viral control of mitochondrial quality and content has emerged as an important mechanism for counteracting the host response to virus infection. Despite the knowledge of this crucial function of some viruses, little is known about how herpesviruses regulate mitochondrial homeostasis during infection. Human herpesvirus 8 (HHV-8) is an oncogenic virus causally related to AIDS-associated malignancies. Here, we show that HHV-8-encoded viral interferon regulatory factor 1 (vIRF-1) promotes mitochondrial clearance by activating mitophagy to support virus replication. Genetic interference with vIRF-1 expression or targeting to the mitochondria inhibits HHV-8 replication-induced mitophagy and leads to an accumulation of mitochondria. Moreover, vIRF-1 binds directly to a mitophagy receptor, NIX, on the mitochondria and activates NIX-mediated mitophagy to promote mitochondrial clearance. Genetic and pharmacological interruption of vIRF-1/NIX-activated mitophagy inhibits HHV-8 productive replication. Our findings uncover an essential role of vIRF-1 in mitophagy activation and promotion of HHV-8 lytic replication via this mechanism.

[1] Department of Oncology, Sidney Kimmel Comprehensive Cancer Center, Johns Hopkins University School of Medicine, Baltimore, MD 21287, USA. [2] Department of Cell Biology, Institute for Basic Biomedical Sciences, Johns Hopkins University School of Medicine, Baltimore, MD 21205, USA. Correspondence and requests for materials should be addressed to Y.B.C. (email: ychoi15@jhmi.edu)

Mitochondria are vital energy-generating organelles in eukaryotic cells. They also integrate signals controlling multiple cellular functions such as apoptosis, cell cycle, and development. In addition, recent studies have shown that mitochondria serve as a platform for mediating antiviral signaling pathways that lead to apoptosis and innate immune responses to virus infection[1,2]. For example, proapoptotic BH3-only proteins (BOPs) are elevated[3] and/or activated during virus replication and induce mitochondrial outer membrane permeabilization, a crucial step in the intrinsic apoptotic process that triggers the release of soluble apoptogenic factors from the intermembrane space. In the innate immune response to virus infection, the RIG-I-like receptors (RLRs) RIG-I and MDA-5 recognize cytosolic viral RNA and promote the oligomerization of the mitochondrial antiviral signaling adaptor protein (MAVS; also known as IPS-1, VISA, and Cardif)[4], which recruits TBK1 and IKKi kinases to activate IRF3 and IRF7 transcription factors[2]. These activated IRFs induce the expression of type I interferon (IFN) genes, the products of which restrict virus replication. Therefore, successful virus infection and replication are in large part achieved by the ability of viruses to attenuate the innate antiviral responses mediated by mitochondria.

Particular viral proteins have evolved to enable virus infection and replication by modulating mitochondria-mediated antiviral responses. For example, human herpesviruses encode anti-apoptotic proteins that inhibit the intrinsic apoptosis pathway[1], and hepatitis C virus encodes a serine protease NS3/NS4A that disrupts RLR signaling and IFN-β production by cleaving MAVS[5]. On the other hand, mitochondria generate reactive oxygen species (ROS) as a byproduct of electron transport complexes. Accumulation of damaged or dysfunctional mitochondria leads to aberrant ROS generation, which can exacerbate apoptosis and augment RLR-MAVS signaling[6,7]. Several viruses are known to effect elimination of infection-altered mitochondria via mitophagy, potentially attenuating apoptosis and innate immune responses[8].

Mitophagy is a key mechanism of mitochondrial quality control that eliminates aged, dysfunctional, damaged or excessive mitochondria via selective autophagy[9,10]. A recent study indicates that impaired removal of damaged mitochondria by mitophagy can lead to stimulator of IFN gene (STING)-dependent inflammation and neurodegeneration[11]. Mitophagy initiated by the mitochondria-localized kinase PINK1 and ubiquitin ligase PARK2 is a well characterized mechanism of mitophagy. After the loss of mitochondrial membrane potential, PINK1 is stabilized on the outer mitochondrial membrane (OMM), where it phosphorylates both ubiquitin (Ub) and the Ub-like domain of PARK2, leading to its activation[12–14]. In turn, activated PARK2 ubiquitinates mitochondrial proteins such as MFN1/2, DRP1, BCL-2, and VDAC1. Mitophagy receptors such as NDP52 and OPTN bind to and link the ubiquitinated mitochondria to autophagosomes via interaction with autophagy-related protein 8 (ATG8) family members such as LC3 and GABARAP[15]. PARK2-independent mitophagy pathways mediated by other mitophagy receptors such as NIX/BNIP3L, BNIP3, FUNDC1, and PHB-2 have been identified under certain physiological conditions[16–19]. Nonetheless, viral regulation of mitophagy is an understudied area of investigation.

Human herpesvirus 8 (HHV-8) is a pathogen associated with endothelial Kaposi's sarcoma (KS) and B-cell diseases primary effusion lymphoma (PEL) and multicentric Castleman's disease[20,21]. HHV-8 productive replication, in addition to latency, is important for maintaining viral load within the host and for KS pathogenesis. Successful HHV-8 replication is in large part achieved by the ability of the virus to inhibit apoptosis and innate immune responses elicited by infection of host cells. HHV-8 encodes a number of proteins expressed during the lytic cycle that have demonstrated or potential abilities to promote virus productive replication via inhibition of antiviral responses activated by infection-induced stress. Among them, viral IFN regulatory factor 1 (vIRF-1) is believed to play important roles in blocking IFN and other stress responses to virus infection and replication through inhibitory interactions with cellular stress signaling proteins such as p53, ATM, IRF-1, GRIM19, and SMAD3/4[22,23]. Furthermore, we found that vIRF-1 localizes to mitochondria upon virus replication and suppresses mitochondria-mediated apoptosis and innate immune responses via its inhibitory interactions with proapoptotic BOPs and MAVS[3,24,25]. However, the precise role of mitochondria-localized vIRF-1 in HHV-8 biology remains elusive. In this study, we identify a role of vIRF-1 in its activation of mitophagy and support of virus replication via this pathway.

## Results

**Mitochondria content decreases during lytic reactivation.** We previously reported that the level of MAVS is significantly diminished in an HHV-8-infected PEL cell line, BCBL-1, in which vIRF-1 expression is induced following lytic reactivation[25]. Another mitochondrial protein TOM20 was also strongly reduced in vIRF-1-expressing lytic cells compared to vIRF-1-negative neighboring cells (Supplementary Fig. 1a). Moreover, MitoTracker Red staining of mitochondria showed an apparent decrease in the levels of mitochondria of vIRF-1-expressing lytic cells (Supplementary Fig. 1b), implying that the reduced levels of MAVS and TOM20 might result from a loss of mitochondria content rather than a specific inhibition of the expression or stability of the proteins. This was examined further by assessing the levels of mitochondrial DNA (mtDNA) using a combined approach of immunofluorescence assay (IFA) and fluorescent in situ hybridization (FISH). Indeed, the result showed that mtDNA was less readily detected in vIRF-1-expressing lytic cells (Fig. 1a). Furthermore, immunoblotting and image cytometry analyses showed that the expression levels of MTCO2 (*mt*DNA-encoded *c*ytochrome c *o*xidase *II*) was significantly reduced in lytic TRExBCBL-1-RTA (hereafter simply termed iBCBL-1) cells, doxycycline (Dox)-inducible for lytic switch protein RTA[26], that were treated with Dox for 2 days (Fig. 1b, c). In addition, immunoblot- and flow cytometry-detected expression levels of TOM20 decreased following lytic reactivation (Fig. 1b, c). Taken together, these results suggest that mitochondria content of cells is downregulated following lytic reactivation of HHV-8.

HHV-8-infected cells undergo apoptosis during lytic infection[27]. Thus, apoptosis might be the cause of the decrease of mitochondria content. To test this, we examined apoptosis in vIRF-1-positive and -negative lytic iBCBL-1 cells that were treated with Dox for 2 days. Terminal deoxynucleotidyl transferase dUTP nick end labeling (TUNEL) assay was employed to detect apoptosis. The results showed that most of TUNEL-positive apoptotic cells were vIRF-1-negative (97%) (Fig. 1d), suggesting that a decrease in mitochondria content in vIRF-1-positive lytic cells is unlikely to be due to increased apoptosis. Treatment of PEL cultures with the pan-caspase and apoptosis inhibitor zVAD did not prevent a lytic replication-induced decrease in mitochondria content while completely inhibiting PARP cleavage, an indicator of apoptosis (Fig. 1e). These results suggest that viral regulation of mitochondria content occurs at an early stage of virus replication that precedes apoptosis.

**vIRF-1 controls mitochondria content via mitophagy.** To examine whether vIRF-1 is involved in the regulation of mitochondria content, we generated a cell line that stably expresses a

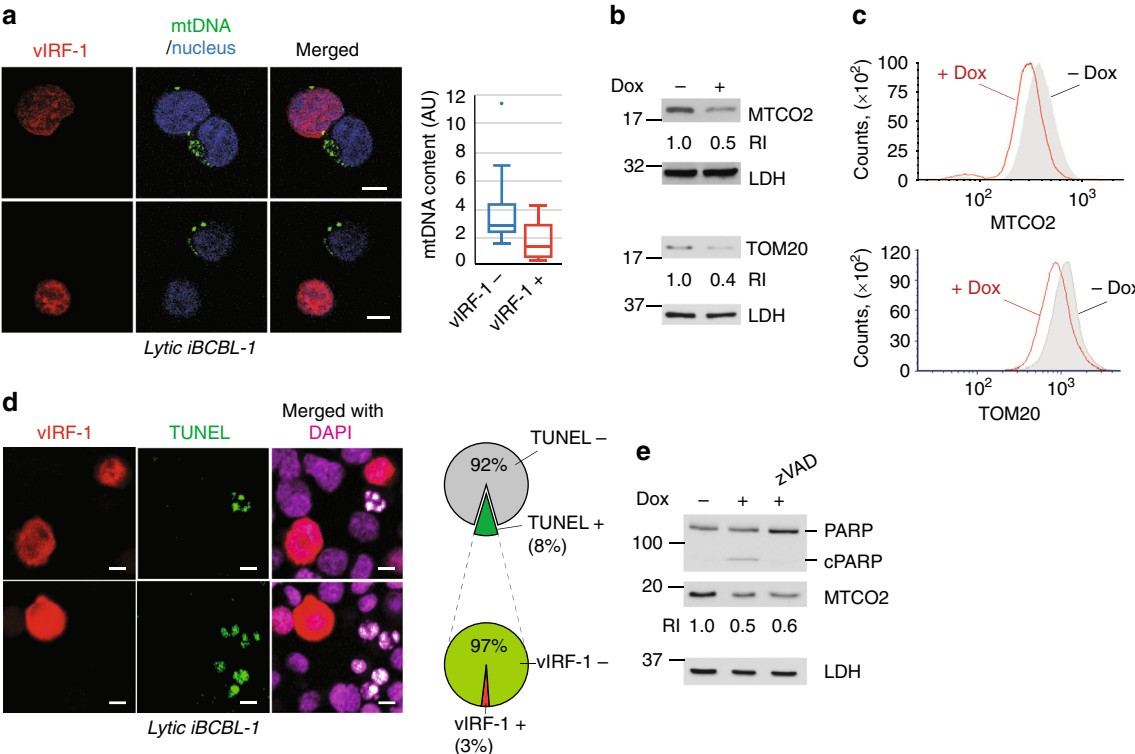

**Fig. 1** Mitochondria content decreases during virus replication. **a** Quantitative assessment of mitochondrial DNA (mtDNA) in lytic (Dox+, 2 days) iBCBL-1 cells; a combined approach of IFA (for vIRF-1 protein) and FISH (for mtDNA) was used. Sixteen images were taken randomly, and representative images are presented. Scale bar, 5 μm. The levels of mtDNA in individual cells were measured using ImageJ, and the distribution of data is shown in the boxplots that display the full range of variation (whiskers), the likely range of variation (box), the median (center line), and the outlier (dot). AU, arbitrary units. **b**, **c** Immunoblotting and image cytometric analyses of MTCO2 and TOM20. Lactate dehydrogenase (LDH) was used as loading control. RI, relative intensities of protein bands, normalized to LDH. **d** In situ detection of apoptosis by TUNEL in lytically reactivated iBCBL-1 cells (Dox+, 2 days). Following TUNEL reaction, the cells were immunostained with vIRF-1 antibody. Approximately 1200 cells were counted from six randomly selected images, and the percentages of TUNEL-positive cells and vIRF-1-negative and -positive cells among TUNEL-positive apoptotic cells are shown in pie charts. Scale bar, 5 μm. **e** PARP cleavage analysis by immunoblotting of the extracts derived from latent and lytic (Dox+, 2 days) iBCBL-1 cells. Ten micromolar zVAD was added to the culture 24 h before cell harvest. Source data are provided as a Source Data file

short-hairpin RNA (shRNA) against vIRF-1 in iBCBL-1 cells along with a control cell line expressing shRNA against luciferase (Luc). Constitutive knockdown of vIRF-1 was previously shown to induce death of latently infected PEL cell lines[28]. To circumvent this limitation, we used an inducible system; Dox-inducible vIRF-1 shRNAs, sh3 and sh5, resulted in 10- to 20-fold decreases in vIRF-1 protein expression compared to control shRNA (sh-Luc) in lytic iBCBL-1 cells (Fig. 2a). We next assessed mitochondria content in the cell lines. Immunoblotting analysis revealed that the relative levels of MTCO2 protein increased in lytic vIRF-1-depleted cells compared to lytic control cells (Fig. 2a). Furthermore, image cytometry analysis showed that, when vIRF-1 was depleted, lytic cells with higher levels of MTCO2 increased in number (Fig. 2b). Taken together, the results suggest that vIRF-1 plays an important role in the downregulation of mitochondria content during virus replication.

The mitochondria content of cells is fundamentally controlled by the balance of anabolic biogenesis and catabolic degradation such as selective autophagy of mitochondria (termed mitophagy). Mitochondrial biogenesis is a complex process requiring the coordinated expression of nuclear and mitochondrial DNAs. Peroxisome proliferator-activated receptor γ-coactivator 1-α (PGC1-α) activates transcriptional cascades involving genes controlling mitochondria biogenesis. One of the genes is the mitochondrial transcription factor A (TFAM) that plays an essential role for transcription and replication of mtDNA, which are important steps in mitochondrial biogenesis[29]. Thus, we

examined the mRNA levels of TFAM but observed no significant difference between control and vIRF-1-depleted iBCBL-1 cells that were left untreated or treated with Dox (Fig. 2c), suggesting that virus replication and vIRF-1 might not influence the transcriptional activation of TFAM for mitochondrial biogenesis. Nonetheless, the level of TFAM protein was highly elevated in vIRF-1-depleted lytic cells but reduced in control cells (Fig. 2d). Therefore, we extrapolated that mitophagy may be involved in vIRF-1 regulation of mitochondria content during lytic replication. To test this notion, we first examined whether HHV-8 activates mitophagy following lytic reactivation. The results of experiments using autophagy inhibitors bafilomycin A1 (Baf A1) and leupeptin showed that these blocked the decrease in the MTCO2 levels in Dox-treated (lytically reactivated) iBCBL-1 cultures (Supplementary Fig. 1c, d). Consistent with this, immunoblotting analysis showed that the decrease of MTCO2 protein was inhibited by Baf A1 and chloroquine (CQ), another autophagy inhibitor, but not by proteasome inhibitor MG132, in Dox-induced iBCBL-1 cultures (Supplementary Fig. 1e). We further examined the formation of mitochondria-containing autolysosomes (hereafter referred to as mitolysosomes), an end-point readout of mitophagy[30], using CellLight[TM] BacMam-labeling of mitochondria and lysosomes (see Methods for details). The results showed that the presence of mitolysosomes was more evident in lytic control iBCBL-1 cells than in latent control cells and both latent and lytic vIRF-1-depleted iBCBL-1 cells (Fig. 2e). Furthermore, electron microscopy imaging demonstrated the

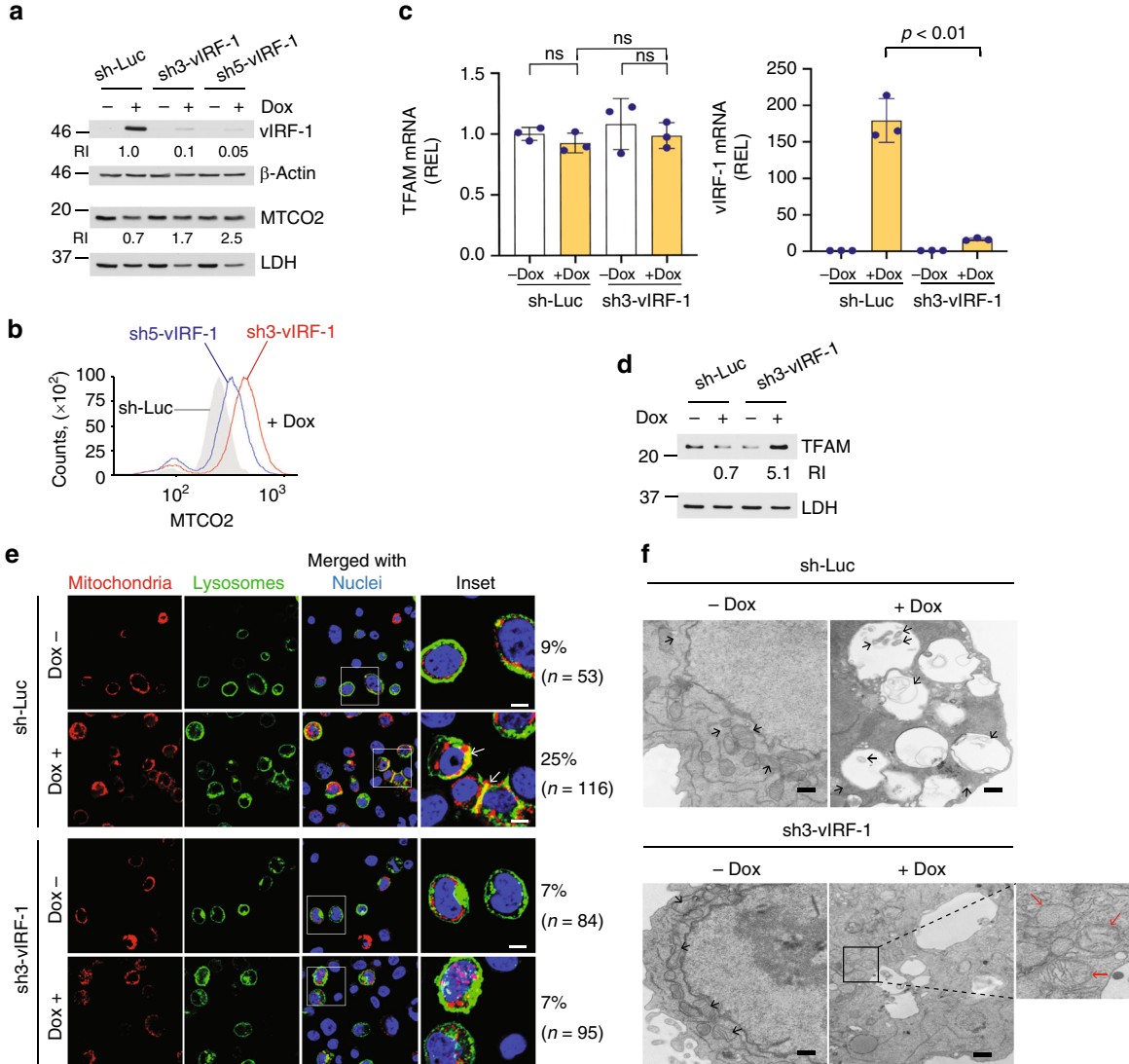

**Fig. 2** vIRF-1 controls mitochondria content via mitophagy. **a** Immunoblot verifications of vIRF-1 silencing by Dox-inducible shRNAs (sh3-vIRF-1 or sh5-vIRF-1). Luciferase shRNA (sh-Luc) was used as control. **b** Image cytometric analysis of MTCO2 expression in vIRF-1-depleted cells versus controls. **c** RT-qPCR analysis of TFAM and vIRF-1 mRNA expression. The relative expression levels (REL) were calculated using comparative Ct ($2^{-\Delta\Delta Ct}$) method. Mean ± SD, $n = 3$. ns, not significant. The $p$-value was determined by matched pair $t$-test. **d** Immunoblot analysis of TFAM protein. The RIs of TFAM are shown below each blot panel. **e** Assessment of mitolysosome formation using CellLight® BacMam 2.0 system (see Methods for details). Representative images are presented, and the yellow areas (white arrows) indicate mitolysosomes. The percentages of mitolysosome-containing cells were determined among reporter-infected cells ($n$) and are noted on the side of the enlarged inset area. Scale bar, 5 μm. **f** Electron microscopy images of the iBCBL-1 cells. Black and red arrows indicate mitochondria with normal and disrupted cristae, respectively. Scale bar, 500 nm. Source data are provided as a Source Data file

presence of mitolysosome-like structures in lytic control cells but not in lytic vIRF-1-depleted iBCBL-1 cells (Fig. 2f). It is noteworthy that mitochondria with disrupted cristae were often observed outside the autophagic vacuoles of lytic vIRF-1-depleted iBCBL-1 cells (Fig. 2f, red arrows). Taken together, our results suggest that vIRF-1 is likely to be involved in activation of mitophagy, thereby controlling mitochondria content of cells during virus replication.

**vIRF-1 activates NIX-mediated mitophagy.** Mitophagy is triggered by activation of specific autophagy receptors localized mainly on the outer mitochondrial membrane (OMM); these proteins interact with ATG8 family members, including LC3 and GABARAP via a short-linear motif termed the ATG8-interacting motif (AIM) or LC3-interacting region (LIR), which forms a

bridge linking the mitochondria to the autophagosomes[10]. Thus, we hypothesized that vIRF-1 may promote mitophagy by recruiting the mitophagy machinery and/or activating it on the mitochondria. Firstly, we investigated changes in the levels of mitophagy proteins, including mitophagy receptors and LC3, on the mitochondria isolated from latent and lytic iBCBL-1 cells. Consistent with our previous report[25], vIRF-1 was readily detected in the mitochondrial fraction isolated from lytic iBCBL-1 cells (Fig. 3a). When autophagy is induced, LC3 is processed from a cytosolic form, LC3-I (18 kDa), to the LC3-II (16 kDa) form that is lipidated with phosphatidylethanolamine and associated with the autophagic vesicle membranes. Intriguingly, the LC3B-II form, but not the LC3B-I form, was readily detected in the mitochondria and here exhibited a more than twofold increase upon virus replication while total LC3B levels remained unchanged after lytic reactivation (Fig. 3a), indicating that

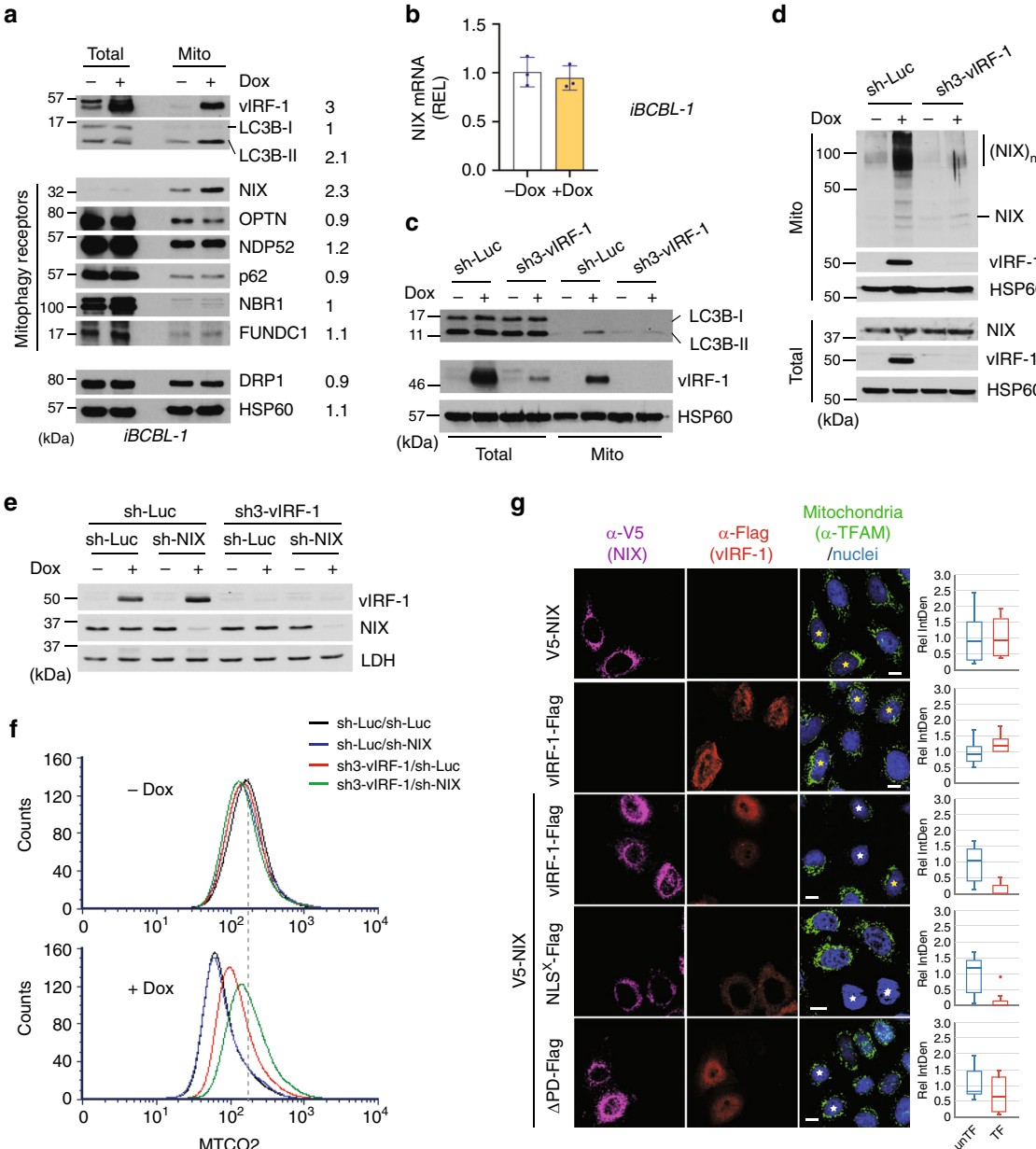

**Fig. 3** vIRF-1 activates NIX-mediated mitophagy. **a** Immunoblots of total-cell and mitochondrial extracts derived from iBCBL-1 cells that were left untreated or treated with Dox for 2 days. The relative band intensities of the Dox-treated mitochondrial fraction to the no Dox control mitochondrial fraction were calculated and are shown in the column (right). Heat shock protein 60 (HSP60) was used as loading control. **b** RT-qPCR analysis of the mRNA expression of NIX in latent and lytic iBCBL-1 cells. The relative expression levels (REL) were calculated using the comparative Ct method. Mean ± SD, $n = 3$. **c, d** Immunoblots of total-cell and mitochondrial extracts derived from control and vIRF-1-depleted iBCBL-1 cells that were left untreated or treated with Dox for 2 days. $(NIX)_n$ indicates a putative dimerized or polymerized form of NIX. **e** Immunoblot validation of Dox-induced depletion of NIX and/or vIRF-1 in the respective shRNA-transduced iBCBL-1 cell lines (see Methods). The cells were left untreated or treated with Dox for 2 days. LDH was used as loading control. **f** Flow cytometric analysis of MTCO2 expression in latent and lytic iBCBL-1 cells that were depleted of NIX and/or vIRF-1. The cells were left untreated or treated with Dox for 4 days. The dotted line shows the peak for MTCO2 in latent cells (no Dox) doubly transduced with sh-Luc. **g** IFA analysis of mitochondria content (TFAM) in HeLa.Kyoto cells that were transiently transfected for 24 h with V5-NIX and/or Flag-vIRF-1 (wild type and its derivatives, NLS$^X$ and ΔPD). Singly and dually transfected cells are marked with yellow and white stars, respectively. TFAM immunofluorescence intensity was measured using ImageJ software; background-corrected integrated density (IntDen) was measured in randomly selected cells ($n = 20$, untransfected (unTF) or transfected (TF) cells) from at least four to five microscope images. Relative IntDen was calculated by dividing by unTF control, and the distribution of data is shown in the boxplots that display the full range of variation (whiskers), the likely range of variation (box), the median (center line), and the outlier (dot). Scale bar, 10 μm. Source data are provided as a Source Data file

selective autophagy of mitochondria is induced during virus replication. Examination of the levels of mitophagy receptors NIX (also termed BNIP3L), OPTN, NDP52, p62, NBR1, and FUNDC1 revealed that the level of mitochondria-associated NIX was increased by more than twofold while the other receptors remained essentially unchanged (Fig. 3a). The levels of the mitochondrial fission protein DRP1 and the mitochondrial chaperone HSP60 remained unchanged (Fig. 3a). *NIX* mRNA

expression was not induced by lytic reactivation (Fig. 3b), indicating that NIX protein may be stabilized and/or translocated to the mitochondria during HHV-8 replication. We failed to detect expression of another mitophagy receptor BNIP3, a paralog of NIX, in both latent and lytic iBCBL-1 cells by immunoblotting using the anti-BNIP3 antibody that was able to readily recognize overexpressed BNIP3 in 293T cells (Supplementary Fig. 2), indicating that BNIP3 may be expressed at low levels in BCBL-1 cells. At present we cannot rule out roles of mitophagy receptors other than NIX in virus replication-activated mitophagy.

Based on our positive data for NIX, we next examined whether vIRF-1 is required for the recruitment of LC3B-II and NIX to mitochondria. Immunoblotting analyses showed that virus replication-induced translocation of LC3B-II to mitochondria was abated when vIRF-1 was depleted (Fig. 3c). Similarly, virus replication-promoted mitochondrial localization of NIX and a shift in apparent molecular mass to about 100-kDa, which may represent a homodimeric form of NIX, were significantly reduced in lytic vIRF-1-deficient iBCBL-1 cells (Fig. 3d). Whether NIX dimerization is required for mitophagy activation remains unclear; however, hypoxia-induced dimerization of a paralog of NIX, BNIP3, was proposed to be related to mitophagy activation in vivo[18]. Therefore, we hypothesized that vIRF-1 may activate NIX-mediated mitophagy via promotion of NIX recruitment to and dimerization on mitochondria. To test this possibility, we transiently transfected HeLa.Kyoto cells with Flag-tagged vIRF-1 (vIRF-1-Flag) and/or V5-tagged NIX (V5-NIX). Contrary to our expectation, there was no effect of vIRF-1 on promotion of V5-NIX recruitment to and dimerization on mitochondria (Supplementary Fig. 3). To examine whether NIX is involved functionally in vIRF-1 regulation of mitochondria content, we stably transduced control and vIRF-1-depleted iBCBL-1 cells with Dox-inducible NIX shRNA (sh-NIX). Dox-inducible depletion of endogenous NIX and vIRF-1 proteins were verified using immunoblotting in each cell line (Fig. 3e). Flow cytometric analysis showed that depletion of NIX alone had a minor effect on the mitochondria content, but double depletion of NIX and vIRF-1 significantly increased mitochondria content in lytic cells to an even greater extent than depletion of vIRF-1 (Fig. 3f), indicative of a mechanistic link between vIRF-1 and NIX. This functional interaction was further assessed by transient transfection experiments; overexpression of both vIRF-1-Flag and V5-NIX, but not each individually, could induce a significant decrease in mitochondria content, as determined by TFAM immunostaining (Fig. 3g). The decrease of mitochondria content was inhibited by autophagy/mitophagy inhibitors, including leupeptin (a lysosomal protease inhibitor), liensinine (an inhibitor of autophagosome-lysosome fusion), and Mdivi-1 (a mitochondrial division inhibitor) (Supplementary Fig. 4), suggesting strongly that vIRF-1 and NIX together induce mitochondrial clearance by activating mitophagy. To rule out a role of nuclear-localized vIRF-1 in the mitochondrial clearance, we used the nuclear localization signal (NLS)-mutated version of vIRF-1 (NLS$^X$) as previously described[24,25]. The vIRF-1 NLS$^X$ variant strongly induced mitochondrial clearance together with V5-NIX (Fig. 3g). However, the vIRF-1 mutant (ΔPD) lacking the proline-rich domain (PD) that contains an atypical mitochondrial targeting signal sequence (MTS)[25] did not significantly induce mitochondrial clearance (Fig. 3g). These results suggest that mitochondrial localization of vIRF-1 is required for mitochondrial clearance. Since overexpression of NIX alone did not induce mitochondrial clearance (Fig. 3g), NIX may require an activation step to function as a mitophagy receptor. For example, it is known that in response to high-oxidative phosphorylation activity, the small GTPase RHEB (Ras homolog enriched in brain) is recruited to the mitochondria and interacts with NIX to

induce mitophagy[31]. Likewise, we hypothesize that mitochondria-localized vIRF-1 may activate NIX-mediated mitophagy by interacting with NIX.

**vIRF-1 binds directly to mitochondria-associated NIX**. To examine whether vIRF-1 interacts with NIX on the mitochondria, we performed a co-immunoprecipitation assay with the mitochondrial fraction isolated from lytic iBCBL-1 cells. The result showed that vIRF-1 was indeed co-precipitated with NIX (Fig. 4a). In addition, IFA analyses showed that vIRF-1 co-localized with NIX in lytic BCBL-1 and transfected HeLa.Kyoto cells (Fig. 4b, c). It is noteworthy that NIX was detected as a highly aggregated form in vIRF-1-positive lytic BCBL-1 cells (white arrows in Fig. 4b). Furthermore, we employed a proximity ligation assay (PLA Duolink, Sigma) to analyze in situ interaction between vIRF-1 and NIX; PLA-derived immunofluorescence signal, indicative of vIRF-1-NIX interaction, was detected in the cytoplasm, potentially in mitochondria, of HeLa.Kyoto cells co-transfected with vIRF-1 and NIX, but not with each alone (Fig. 4d). To examine whether vIRF-1 binds directly to NIX, we performed an in vitro GST-pull-down assay using purified recombinant proteins. Recombinant vIRF-1 protein, which was purified from bacteria[25], was co-precipitated with purified GST-NIX but not GST protein (Fig. 4e), and recombinant NIX protein (Sino Biological) was co-precipitated with purified GST-vIRF-1 but not GST (Fig. 4f).

To further characterize this interaction, we next sought to identify the vIRF-1 region(s) responsible for NIX binding. In an in vitro binding assay, we used purified GST-NIX and cell lysates of 293T cells transfected with vIRF-1 full-length (FL) or a series of deletion variants in which segments of 23 to 25 residues were deleted (Fig. 4g). Deletion of the first segment encompassing residues 1 to 23, comprising part of the PD region, completely abrogated NIX binding of vIRF-1 (Fig. 4g). Subsequent mutagenesis studies were carried out to identify vIRF-1 residues required for NIX binding (Fig. 4g). These experiments identified two residues of vIRF-1, asparagine 8 (N8) and phenylalanine 10 (F10), as essential for NIX binding (Fig. 4g). Furthermore, the F10A mutation led to a loss of the ability of vIRF-1 to induce mitochondrial clearance together with NIX (Fig. 4h), showing that vIRF-1 interaction with NIX is required to activate NIX-mediated mitophagy. Collectively, our results indicate that the N-terminal PD region contains motifs for vIRF-1 binding to NIX and activation of NIX-mediated mitophagy and for mitochondrial targeting[25].

**NIX requirements for interaction with vIRF-1**. To characterize the intracellular interaction of vIRF-1 and NIX, we employed a luciferase-based protein fragment complementation assay, NanoLuc® Binary Technology (NanoBiT, Promega). The Nano-BiT system consists of two small subunits, Large BiT (LgB; 17.6 kDa) and Small BiT (SmB; 1.3 kDa) (Fig. 5a). To find an optimal orientation of the NanoBiT tags for detection of vIRF-1:NIX interaction, Flag-vIRF-1 and V5-NIX were tagged at either the N- or C-terminal end of the NanoBiT subunits (Supplementary Fig. 5a). Intriguingly, when the NanoBiT tags were fused to the C-terminal end of NIX, the formation of a NIX dimer was greatly reduced (Supplementary Fig. 5b); this is likely due to its steric hindrance in the C-terminal tail-anchor (TA) region, thus interfering with targeting to the OMM and/or TA region-mediated dimerization[32]. At 24 h after co-transfection with the different vIRF-1 and NIX plasmid combinations or single plasmids, a Nano luciferase (NLuc) substrate, furimazine, was added to the cell cultures. We identified two binary combinations showing the highest NLuc activity: (1) vIRF-1-SmB plus LgB-NIX

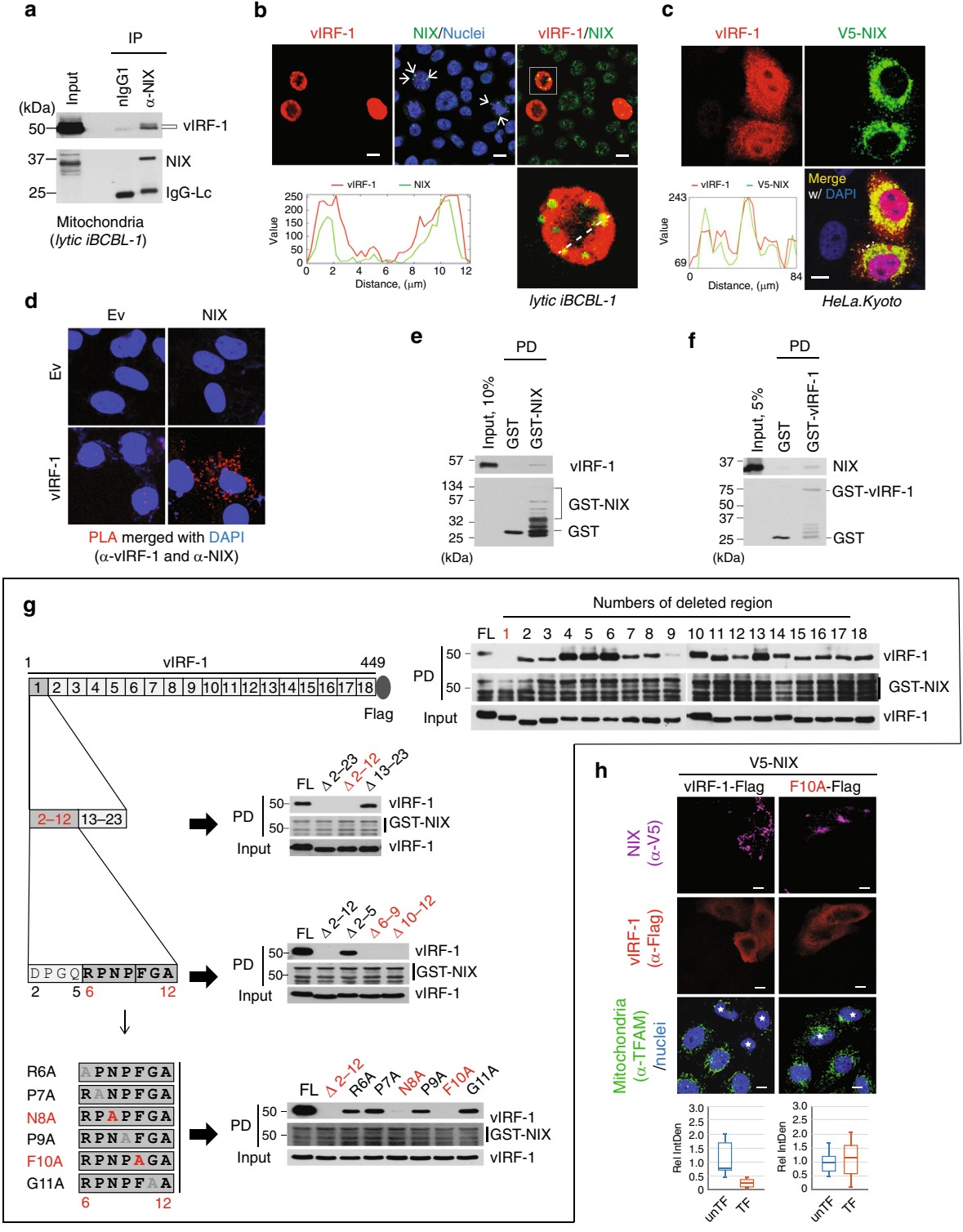

and (2) SmB-vIRF-1 plus LgB-NIX (Supplementary Fig. 5c). To further demonstrate their specific interactions, we generated various deletion and substitution variants together with a negative control, HaloTag (HT)-SmB (Fig. 5b). NIX indeed bound specifically to vIRF-1; the luminescence signal was tenfold higher when LgB-NIX was co-expressed with vIRF-1-SmB than with HT-SmB (Fig. 5c). However, NIX binding to vIRF-1 was significantly diminished when the PD of vIRF-1 was deleted (Fig. 5d). These results are consistent with the data from the precipitation assays above and validated the NanoBiT system for further quantitative

analysis of vIRF-1 and NIX interaction. We next examined whether the mitochondrial targeting of NIX is required for binding to vIRF-1. For this assay, we generated a NIX variant (NIXΔTA) deleted of the tail-anchor (TA) region (residues 188 to 208) and showed that this variant lost the ability to localize to mitochondria and to form a homodimer (Supplementary Fig. 6 and Fig. 5e). The NIX variant also failed to interact with vIRF-1 (Fig. 5f). To further confirm the requirement of NIX mitochondrial localization for vIRF-1 binding, we substituted the TA of NIX with that of other mitochondrial TA proteins, VAMP1B and

**Fig. 4** vIRF-1 interacts physically and functionally with NIX. **a** Co-immunoprecipitation (co-IP) of vIRF-1 and NIX from the mitochondrial fraction derived from lytic iBCBL-1 cells (+Dox for 2 days). Isotype-matched normal mouse IgG1 (nIgG1) was used as control at an equivalent concentration. **b**, **c** IFA analyses of vIRF-1 and NIX proteins in lytic BCBL-1 (**b**) and HeLa.Kyoto cells transfected with vIRF-1 and V5-NIX (**c**). Arrows indicate an aggregate form of NIX in vIRF-1-expressing iBCBL-1 cells. The fluorescence intensity profiles were generated using the segmented line tool and multichannel plot profile command in ImageJ. **d** In situ proximity ligation assay (PLA) of vIRF-1 interaction with NIX in HeLa.Kyoto cells transfected with vIRF-1 and/or NIX. Rabbit anti-vIRF-1 and mouse anti-NIX (E-1) antibodies were used. The red fluorescence indicates PLA-positive signals. **e**, **f** In vitro GST-pull-down (PD) assays of vIRF-1 interaction with NIX. Purified recombinant vIRF-1 and NIX was pulled down with glutathione sepharose beads coated with 1 μg of GST-fusion NIX (**e**) and vIRF-1 (**f**) proteins, respectively. GST alone was used as control. The precipitated complexes were separated by SDS-PAGE and immunoblotted with the indicated antibodies. **g** Identification of the vIRF-1 residues required for NIX binding. Initial screening employed a series of vIRF-1 variants in which each of 18 segments of about 25 residues were deleted; next, the first N-terminal segment was progressively deleted. Finally, individual alanine substitutions of residues 6 to 11 were introduced. GST-pull-down assays were performed with cell extracts of 293T cells expressing vIRF-1 full-length (FL) or variants. The regions or residues found to be required for NIX binding are highlighted in red. **h** IFA analysis of mitochondria content (TFAM) in HeLa.Kyoto cells transiently transfected with V5-NIX and vIRF-1-Flag (wild type or F10A) for 24 h. The double-transfected cells are marked with white stars. Relative IntDen was calculated by dividing by unTF control, and the distribution of data is shown in the boxplots that display the full range of variation (whiskers), the likely range of variation (box), and the median (center line). Scale bar, 5 μm. Source data are provided as a Source Data file

FIS1, and tested these for interaction with vIRF-1. Both chimeric proteins, NIX-TA$^{VAMP1B}$ and NIX-TA$^{FIS1}$, properly localized to mitochondria (Supplementary Fig. 6) and bound vIRF-1 (Fig. 5f).

Unlike NIX, however, immunoblotting analysis showed that the chimeric NIX proteins did not form homodimers (Fig. 5f). Furthermore, NIX-TA$^{VAMP1B}$ did not induce mitochondrial clearance together with vIRF-1 (Fig. 5g). These results indicate that NIX dimerization via its own TA region is essential for mitophagy activation, and the interaction of vIRF-1 and NIX is not sufficient for triggering mitophagy without NIX dimerization. ATG8 binding to the LIR motif of NIX is critical in the NIX-mediated mitophagy pathway[17]. Indeed, we found that a NIX variant (NIXΔLIR) deleted of the LIR motif (residues 35 to 39) did not induce mitochondrial clearance together with vIRF-1 (Fig. 5g). Accordingly, we postulated that vIRF-1 may promote the interaction of NIX with ATG8 family members via the LIR. To test this, we employed a co-IP assay using HA-tagged LC3B (HA-LC3B). However, we could not discern any effect of vIRF-1 on NIX and LC3B interaction, while vIRF-1, but not vIRF-1 F10A, bound to NIX (Fig. 5h). These results suggest that vIRF-1 may activate NIX-mediated mitophagy by promoting NIX interaction with other member(s) of the ATG8 family or via an unknown mechanism(s).

**vIRF-1 promotes survival of lytic cells via mitophagy.** Given that the PD of vIRF-1 contains the sequences for both mitochondrial targeting and NIX binding, we predicted that deletion of the PD may impede virus replication-induced mitophagy. To test this, we generated an HHV-8 bacterial artificial chromosome 16 variant (BAC16.ΔPD) encoding vIRF-1 lacking the PD region, using λ-Red recombination techniques described previously[33]. We verified the deletion of the targeted region and the integrity of the mutated versus wild-type BAC16 DNA by sequencing and gel electrophoresis after AvrII or SpeI digestion (Supplementary Fig. 7). BAC16 (wild type) and BAC16.ΔPD DNAs were stably transfected into iSLK cells, which express Dox-inducible HHV-8 immediate-early protein RTA and provide a tractable model cell line for HHV-8 infection[34]. We first investigated mitochondria content using flow cytometry analysis of MTCO2 and TOM20 in latent and lytically reactivated (sodium butyrate and Dox-treated) iSLK cells. The results showed that the populations with higher MTCO2 and TOM20 (the M2 region of Fig. 6a, b) were increased by about three- to fourfold in cells lytically infected with BAC16.ΔPD compared to those infected with wild-type BAC16. Therefore, these data support our hypothesis that vIRF-1 plays a critical role in the downregulation of mitochondria content via its PD region. In line with this, lentiviral transduction of vIRF-1 into BAC16.ΔPD cells promoted an increase in the TOM20-low

population (the M1 region) and a decrease in the TOM20-high population (M2) (Fig. 6c). Furthermore, we assessed the formation of mitolysosomes in latently and lytically infected BAC16 and BAC16.ΔPD cells by immunostaining of TOM20 and LAMP1, a lysosome marker. We used the lysosomal protease inhibitor leupeptin (Leu) to facilitate the detection of mitolysosomes in the iSLK cells (Fig. 6d). The results showed that the formation of mitolysosomes was evident in lytic BAC16 cells, but not in control (BAC16-negative) and lytic BAC16.ΔPD cells (Fig. 6d, e). Moreover, vIRF-1-transduced BAC16.ΔPD cultures showed a higher percentage of mitolysosome-positive cells during lytic replication compared to empty vector-transduced cultures (Fig. 6f, g). These genetic studies strongly support our hypothesis that mitochondria-targeted vIRF-1 activates mitophagy during HHV-8 lytic replication.

We previously demonstrated that mitochondria-localized vIRF-1 plays an important pro-survival role in lytic iBCBL-1 cells[25]. We thus wanted to investigate this activity of vIRF-1 in the context of BAC16-infected iSLK cells. When reactivated with SB/Dox treatment for 2 days, BAC16 iSLK cells exhibited modest morphological changes (cytopathic effects (CPE)), including rounding of cells, characteristic of apoptosis, as assessed by bright-field microscopy (Fig. 6h). However, BAC16.ΔPD cells showed stronger CPE (Fig. 6h). Furthermore, we observed that apoptosis, as assessed by Annexin-V staining and PARP cleavage, was more apparent in BAC16.ΔPD cells than in BAC16 cells upon reactivation (Fig. 6i, j, respectively). In addition, lytic replication-induced PARP cleavage was inhibited by vIRF-1 transduction into BAC16.ΔPD cells (Fig. 6k). Taken together, these results suggest that mitochondria-localized vIRF-1 mediates, through promotion of mitophagic clearance of mitochondria, protection of lytic cells from mitochondrial-induced cytotoxicity.

**A vIRF-1 PD peptide dysregulates mitochondrial homeostasis.** To test the possibility that a short peptide containing the mitochondrial- and NIX-targeting sequences of vIRF-1 may competitively inhibit vIRF-1-activated mitophagy, we synthesized a peptide (TAT-PD1) comprising vIRF-1 residues 1-25 fused to HIV-1 TAT residues 47-57 (Fig. 7a). The TAT-only peptide (hereafter termed TAT) was used as a control. Firstly, we examined whether TAT-PD1 can target to the mitochondria. For IFA detection of the peptides, we used mouse anti-TAT antibody that recognizes residues 47-58 of HIV-1 TAT. Consistent with our previous report[25], most TAT-PD1 was detected in the mitochondria, stained with prohibitin (PHB) antibody, of lytic iBCBL-1 cells while TAT appeared to be nuclear-localized (Fig. 7b). It is noteworthy that the

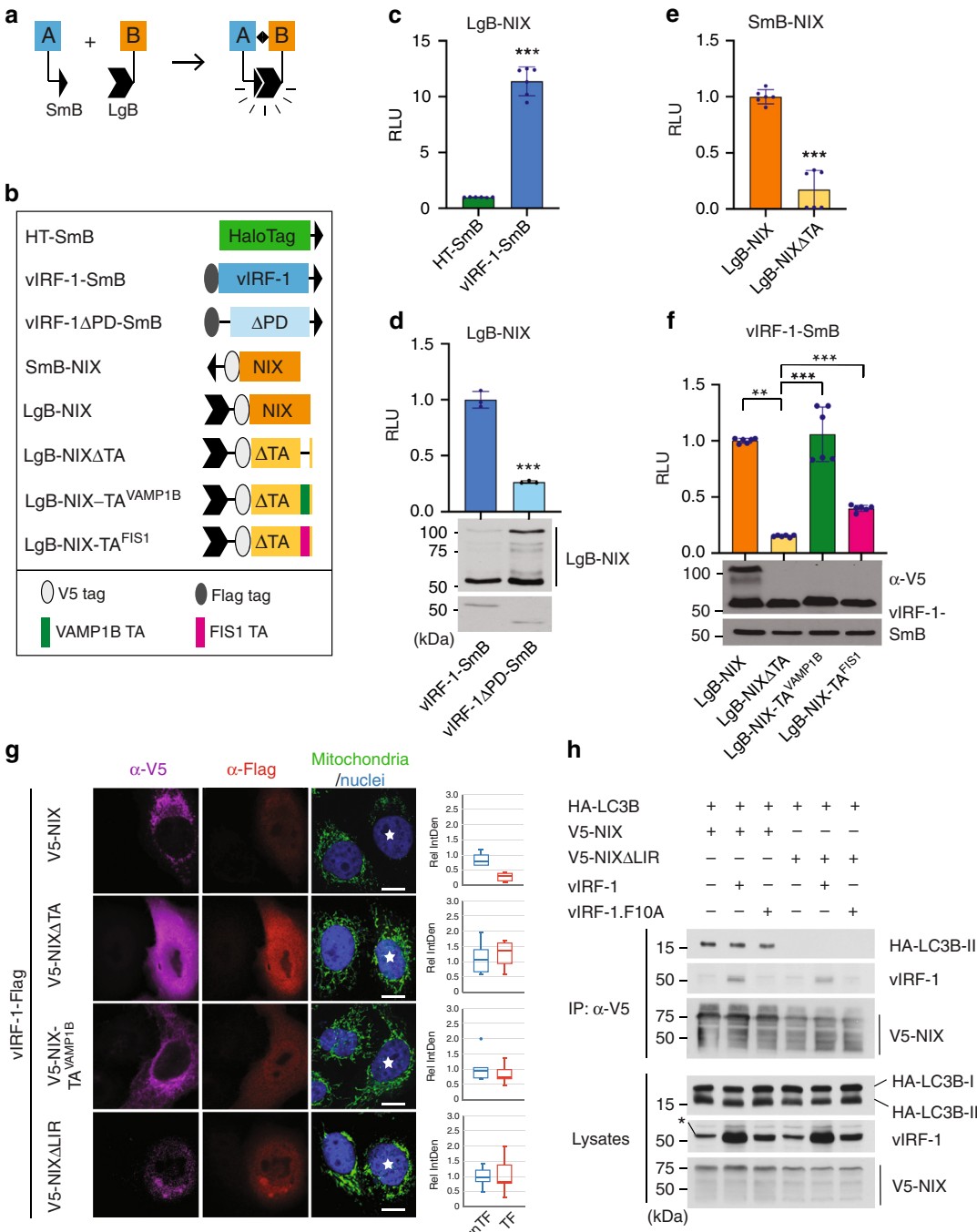

**Fig. 5** Structural requirement of NIX for interaction with vIRF-1. **a** Principle of the NanoBiT-based protein fragment complementation assay (PCA). When a protein [A] binds to its partner [B], their fused Small BiT (SmB; 1.3 kDa) and Large BiT (LgB; 17.6 kDa) fragments are brought into proximity, which allows structural complementation thus yielding a functional enzyme acting on Nano luciferase substrate. **b** Schematic structure of the NanoBiT-fused proteins used in the following studies. TA indicates the tail-anchor domain. **c**–**f** NanoBiT PCA assays were performed using 293T cells transiently transfected with the indicated NanoBiT binary plasmids. Each value represents the mean of triplicate samples from two independent experiments. Error bars represent standard deviations. The p-values were determined by matched pair t-test (**p < 0.01 and ***p < 0.001) **c** The optimal orientation of vIRF-1 and NIX (vIRF-1-SmB and LgB-NIX) was compared to HaloTag (HT)-SmB and LgB-NIX fusion proteins (See also Supplementary Fig. 5). **g** IFA analysis of mitochondria content (TFAM) in HeLa.Kyoto cells that were transiently transfected for 24 h with vectors expressing vIRF-1-Flag and V5-NIX (wild type or variants, NIX-TA^VAMP1B and NIXΔLIR lacking the LC3-interacting region (LIR)). Double-transfected cells are marked with white stars. Relative IntDen was calculated by dividing by unTF control and is shown in a Box-Whisker diagram. Scale bar, 10 μm. **h** Co-IP analysis of the effect of vIRF-1 on NIX interaction with LC3B. HeLa.Kyoto cells transiently transfected with vectors expressing HA-LC3B and V5-NIX wild type or mutant (NIXΔLIR) in the presence or absence of vIRF-1 wild type and F10A mutant. Cells were treated overnight with bafilomycin A1 before their harvest. Asterisk indicates a non-specific band. Source data are provided as a Source Data file

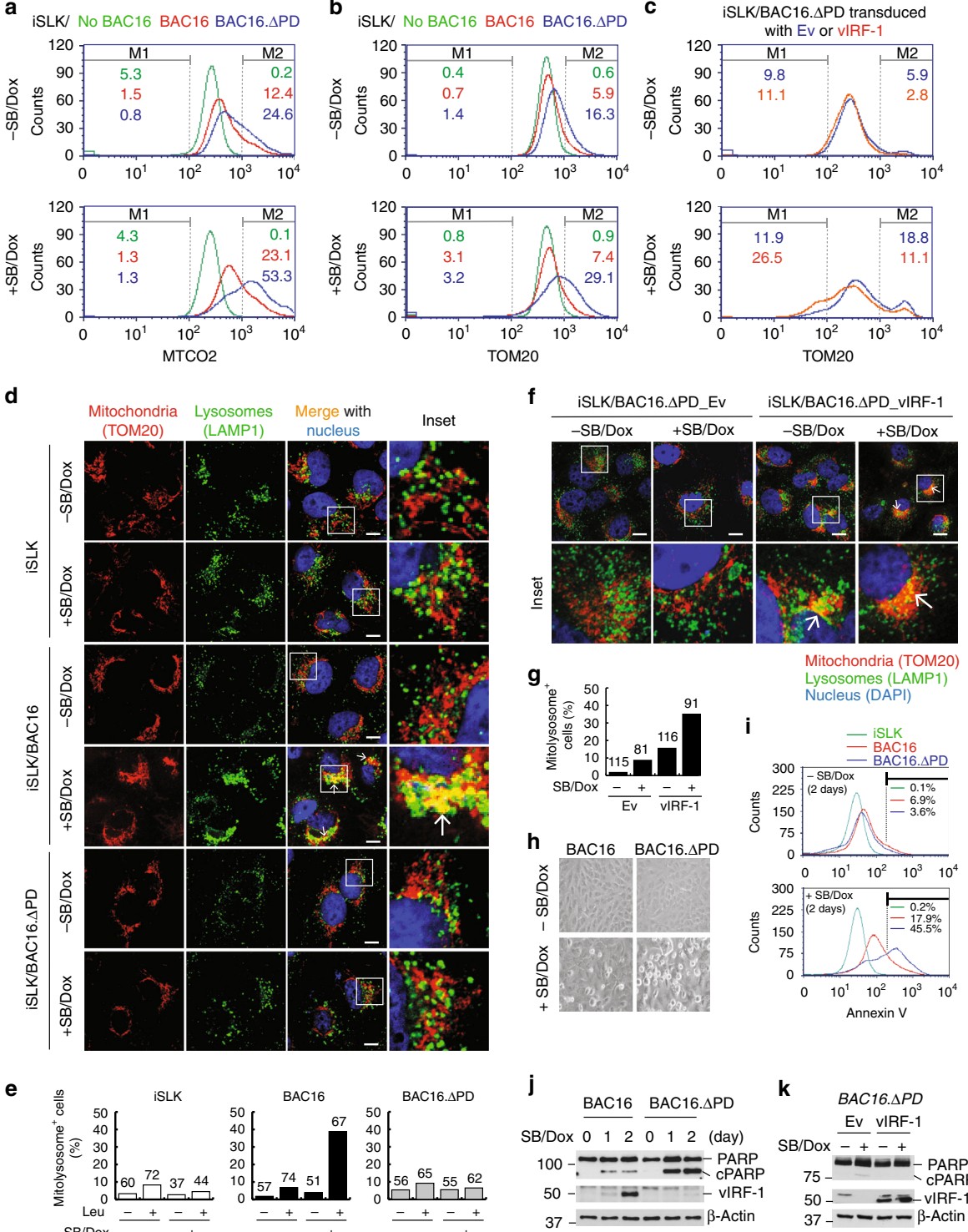

immunofluorescence signal of the mitochondria in lytic iBCBL-1 cells appeared to be more intense when cells were treated with TAT-PD1 (see the magnified images of Fig. 7b). Furthermore, TAT-PD1, but not TAT, could inhibit the mitochondrial clearance induced by vIRF-1 and NIX in HeLa.Kyoto cells and promoted sequestration of the mitochondria at the perinuclear area in the cells (Fig. 7c). Next, we examined whether TAT-PD1 influences mitochondria content in HHV-8-infected cells. Flow cytometric analysis of MTCO2 showed that TAT-PD1, but not TAT, elicited an increase of the cell population with a higher

mitochondria content in lytic iBCBL-1 cells (compare M3 between the right panels of Fig. 7d). Also, TAT-PD1 effectively diminished the cell population with lower mitochondria content (compare M1 between the right panels of Fig. 7d), indicating that TAT-PD1 could promote an accumulation of the mitochondria by blocking mitochondrial clearance in lytic cells. Intriguingly, TAT-PD1 also led to an accumulation of the M3 population in latent cells, but to a lesser extent compared to lytic cells (Fig. 7d, low left panel), implying that TAT-PD1 might interfere with basal mitophagy, presumably mediated by

**Fig. 6** vIRF-1-activated mitophagy is essential for cell survival during virus replication. **a**, **b** Flow cytometric analyses of mitochondria content (MTCO2 and TOM20) of iSLK cells that were uninfected or infected with HHV-8 BAC16 wild type or its variant BAC16.ΔPD that lacks the PD region of vIRF-1. See also Supplementary Fig. 7. The iSLK cells were left untreated or treated with sodium butyrate (SB) and Dox for 2 days. Histogram M1 and M2 regions indicate the populations with lower and higher levels of mitochondria content, respectively, compared to the peak area. **c** BAC16.ΔPD cells were lentivirally transduced with control empty vector (Ev) or vIRF-1 and subjected to flow cytometric analysis as above. **d** IFA analysis of mitolysosomes from iSLK, BAC16, and BAC16.ΔPD cells that were left untreated or treated with SB and Dox for 2 days. To facilitate detection of mitolysosomes, lysosomal protease inhibitor leupeptin (leu) or vehicle control (water) was added to the cultures 8 h before fixation. As BAC16 and BAC16.ΔPD cells express GFP, TOM20 was stained with far-red fluorophore Alexa Fluor 647 and is displayed with pseudo green color allowing the observation of the co-localization with yellow color. Scale bar, 10 μm. **e** Determination of the number of mitolysosome-containing cells from the IFA analysis (**d**). Multiple images were taken randomly of each sample, and the number of cells showing apparent co-localization (i.e., yellow areas indicated by white arrows) of mitochondria (TOM20) and lysosomes (LAMP1) were counted and the derived data are presented graphically. **f**, **g** IFA analysis of mitolysosomes from BAC16.ΔPD cells transduced with Ev or vIRF-1. Scale bar, 10 μm. **h** Phase-contrast images of iSLK BAC16 and BAC16.ΔPD cells untreated and treated with SB and Dox for 2 days. **i–k** Apoptosis assays. Annexin-V staining and PARP cleavage assays were conducted in the iSLK cells that were left untreated or treated with SB and Dox for the indicated period. BAC16.ΔPD cells transduced with Ev or vIRF-1 were reactivated for 2 days with SB and Dox (k). cPARP, cleaved PARP. Source data are provided as a Source Data file

vIRF-1 that is expressed at low levels in latent PEL cells or by a vIRF-1-independent mechanism.

Most strikingly, electron microscopy analysis revealed that TAT-PD1, but not TAT alone, induced a substantial enlargement of the mitochondria in lytic iBCBL-1 cells (Fig. 7e, f). This may be attributed to an accumulation of fused mitochondria and/or inhibition of mitochondrial fission by the peptide, together with concomitant inhibition of mitophagy. Interestingly, overexpression of vIRF-1 induced mitochondrial fragmentation, and TAT-PD1, but not TAT, inhibited vIRF-1-induced mitochondrial fragmentation (Fig. 7g). The fragmentation could be mediated by a block of fusion and/or promotion of fission. To test this, we used a dominant-negative variant (K38A) of DRP1, a mitochondrial fission protein, and found that vIRF-1-induced mitochondrial fragmentation was inhibited by the variant (Fig. 7h), suggesting that vIRF-1 may induce mitochondrial fragmentation . As mitochondria fission is often associated with initiation of mitophagy[35], it is likely that vIRF-1-induced mitochondrial fission may contribute to the promotion of mitophagy although it remains to be determined the exact mechanisms by which vIRF-1 orchestrates mitochondrial dynamics to induce mitophagy. Interestingly, electron microscopy revealed that TAT-PD1 induced nuclear condensation in lytic cells (Fig. 7e), indicative of apoptotic cell death. Supporting this observation, annexin-V staining analysis showed that TAT-PD1 significantly promoted apoptotic cell death of lytic iBCBL-1 cells (Fig. 7i).

**Mitophagy is essential for HHV-8 productive replication.** We previously demonstrated that mitochondria-localized vIRF-1 contributes to HHV-8 productive replication[3,25]. We sought to verify these findings using the Dox-inducible system described above. Consistent with our published data[3,25], production of encapsidated virions as well as protein expression of lytic antigens, including ORF45 and K8.1 decreased by 40 to 50% in lytic vIRF-1-depleted iBCBL-1 cells compared to lytic control (sh-Luc) cells (Fig. 8a, b). Furthermore, the loss of the vIRF-1 function was exacerbated by simultaneous depletion of NIX (Fig. 8a, b). However, depletion of NIX alone had marginal effects on virus productive replication and ORF45 protein expression. These results reflect our mitochondria content studies (Fig. 3f). In contrast to lytic protein expression, corresponding lytic mRNA expression was not affected by vIRF-1 and/or NIX depletion (Fig. 8c). This may be due to differences in susceptibility of transcription and translation to apoptosis; the rate of protein synthesis is rapidly downregulated following induction of apoptosis compared to that of mRNA synthesis[36,37]. Overall, our data indicate that vIRF-1-activated mitophagy promotes virus productive replication by protecting lytic cells from apoptosis.

Lastly, we examined the effect of the mitophagy-inhibitory vIRF-1 peptide on virus productive replication. As expected, TAT-PD1, but not TAT, significantly inhibited the production of encapsidated HHV-8 virions in iBCBL-1 cells that were treated with Dox for 48 and 72 h (Fig. 8d). Interestingly, spontaneous virus production in the absence of Dox was greatly suppressed by TAT-PD1 (Fig. 8d), conceivably due to a cytotoxic effect caused by an accumulation of mitochondria as shown in Fig. 7d. Furthermore, mitophagy inhibitor liensinine (used at 20 μM) inhibited basal and lytic production of HHV-8 virions (Fig. 8e), and mitochondrial fission inhibitor Mdivi-1 also inhibited production of HHV-8 virions from lytic iBCBL-1 cells with a half maximal inhibitory concentration ($IC_{50}$) of about 25 μM (Fig. 8f). In short, our findings demonstrate the importance of mitophagy for productive replication of HHV-8.

## Discussion
The role of autophagy in HHV-8 lytic infection has been controversial. The HHV-8-encoded lytic proteins K7 and vBcl-2 suppress autophagy by interacting with autophagy proteins Rubicon and Beclin 1, respectively[38,39], reportedly promoting virus replication. In addition, other lytic proteins including K1, vIL-6, vGPCR, and ORF45 may inhibit the initiation of autophagy by activating PI3K/AKT/mTOR signaling[40]. However, these studies are sharply in contrast to another report that found that RTA protein, a switch for turning on lytic reactivation, can activate autophagy to facilitate HHV-8 lytic replication[41]. A recent report also showed that autophagy is activated concurrently with HHV-8 lytic induction and is essential for viral lytic gene expression in multiple PEL cell lines[42]. Intriguingly, the authors proposed that the fusion of autophagosomes with lysosomes may be blocked at the late steps of virus replication to allow virus particles to be delivered to the extracellular milieu via autophagic vesicles. These results suggest that autophagy might be subject to differential, stage-specific regulation by HHV-8 during the course of productive replication. Nonetheless, the role of selective autophagy in HHV-8 infection has not previously been reported. Herein, we have provided experimental evidence for selective autophagy of mitochondria (mitophagy) in cells lytically infected with HHV-8, and further demonstrated that mitochondria-localized vIRF-1 activates mitophagy by directly interacting with the mitophagy receptor, NIX, to facilitate HHV-8 replication.

Our experiments with Dox-inducible double silencing of vIRF-1 and NIX in lytically HHV-8-infected PEL cells have demonstrated that vIRF-1 and NIX are mechanistically linked for mitochondrial clearance. Conversely, overexpression of both vIRF-1 and NIX, but not each alone, promoted mitochondrial

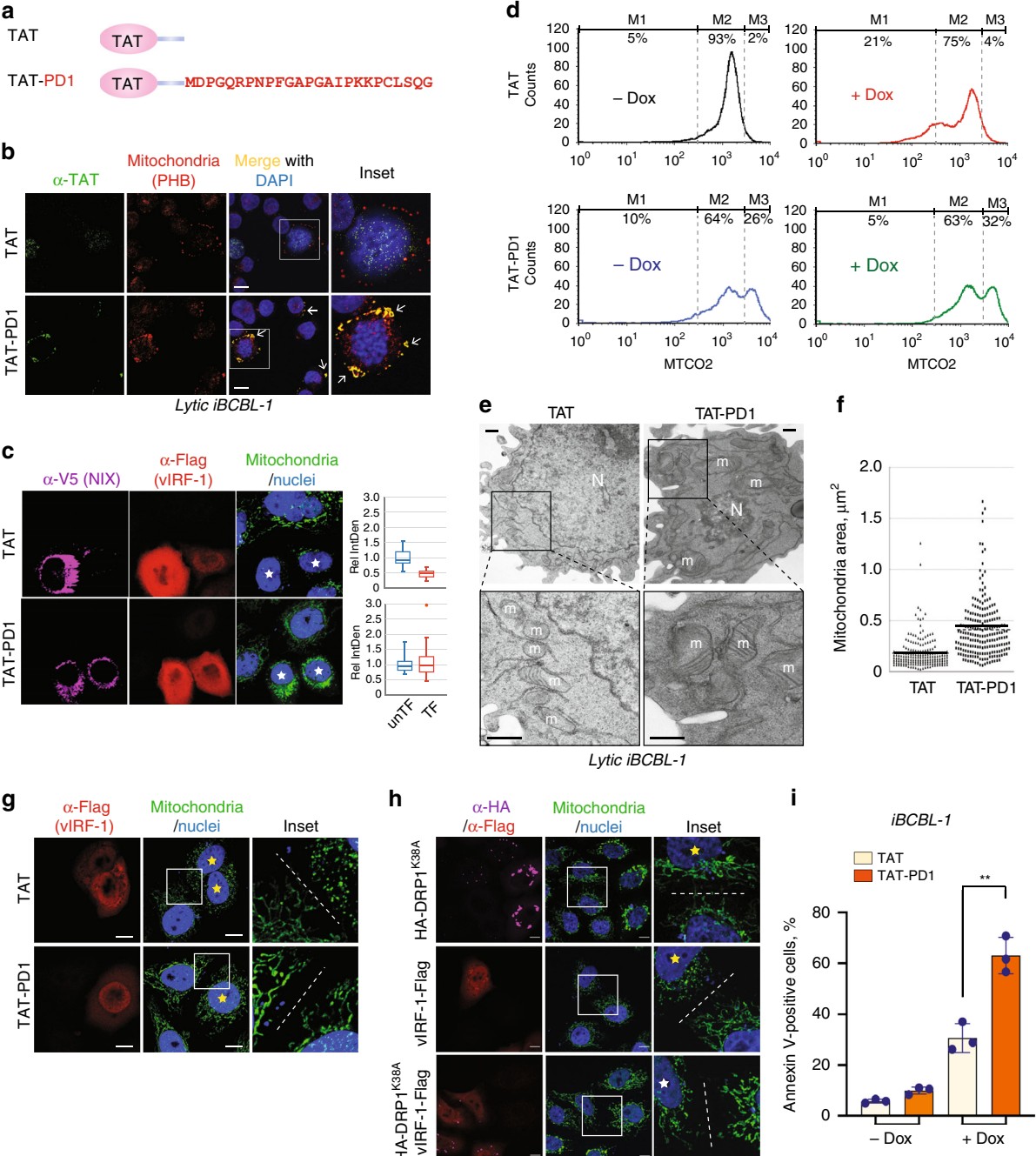

**Fig. 7** A vIRF-1 peptide impairs mitophagy and mitochondrial dynamics. **a** TAT and TAT-fused PD1 (TAT-PD1) peptides. PD1 consists of the N-terminal vIRF-1 residues 1 to 25. **b** IFA analysis of intracellular delivery of the TAT peptides. iBCBL-1 cells were treated with 10 μM TAT or TAT-PD1 peptide together with Dox for 2 days and immunostained with anti-TAT and prohibitin (PHB, mitochondrial marker) antibodies. White arrows indicate co-localization of TAT-PD1 with mitochondria. Scale bar, 10 μm. **c** IFA analysis of TFAM of HeLa.Kyoto cells that were transfected with V5-NIX and vIRF-1-Flag and then treated with 20 μM TAT or TAT-PD1. Relative IntDen was calculated by dividing by unTF control, and the distribution of data is shown in the boxplots that display the full range of variation (whiskers), the likely range of variation (box), the median (center line), and the outlier (dot). **d** Flow cytometric analysis of MTCO2 of iBCBL-1 cells that were treated with 10 μM TAT or TAT-PD1 together with or without Dox for 2 days. **e**, **f** Electron microscopy analysis of lytic iBCBL-1 cells (Dox for 2 days) treated with 10 μM TAT or TAT-PD1. m indicates mitochondria and N indicates nucleus. Scale bar, 500 nm. Mitochondria areas were determined using ImageJ software and are depicted on the dot graph (**f**). The numbers of mitochondria and cells measured: $n = 187$, 20 cells for TAT and $n = 208$, 26 cells for TAT-PD1. **g** IFA analysis of mitochondria (TFAM) of HeLa.Kyoto cells transfected with vIRF-1-Flag and treated with 20 μM TAT or TAT-PD1 for 24 h. Scale bar, 5 μm. **h** IFA analysis of mitochondria (TFAM) of HeLa.Kyoto cells transiently transfected with a dominant negative form of DRP1 (HA-DRP1 K38A) and/or vIRF-1-Flag for 24 h. Singly and dually transfected cells are highlighted in yellow and white stars, respectively. Dotted lines distinguish between transfected and untransfected cells. Scale bar, 5 μm. **i** Flow cytometric analysis of Annexin-V binding to iBCBL-1 cells treated with 10 μM TAT or TAT-PD1 (± Dox for 2 days). Each value represents the mean of three independent experiments ± SD . The $p$-value was determined by matched pair $t$-test (**$p < 0.01$). Source data are provided as a Source Data file

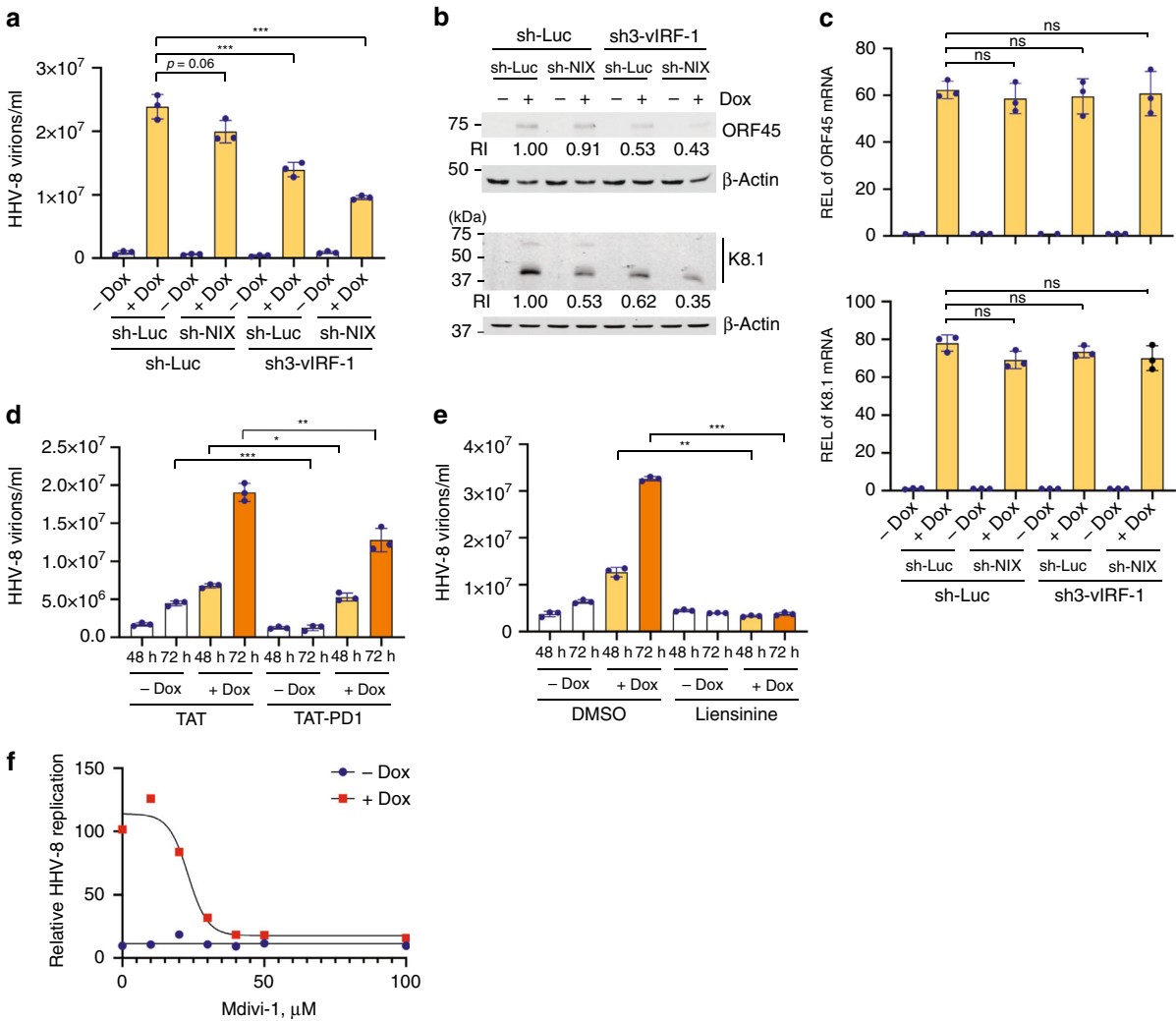

**Fig. 8** Mitophagy plays an essential role in HHV-8 productive replication. **a** RT-qPCR analysis of the copy number of encapsidated viral genome present in the culture media of control (sh-Luc) and vIRF1- and/or NIX-depleted iBCBL-1 cells that were left untreated or treated with Dox for 4 days. Mean ± SD, $n$ = 3. The $p$-value was determined by matched pair $t$-test (***$p$ < 0.001). **b** Immunoblots of total-cell extracts derived from the same cultures as above (**a**) using antibodies to lytic antigens ORF45 and K8.1. The relative band intensities (RI) normalized to the loading control β-Actin are displayed beneath the corresponding panel. **c** RT-qPCR analysis of the mRNA expression of ORF45 and K8.1 in control (sh-Luc) and vIRF1- and/or NIX-depleted iBCBL-1 cells that were left untreated or treated with Dox for 4 days. The relative expression levels (REL) normalized to 18S rRNA was calculated by dividing Dox sample by no-Dox control in each cell line. Data represent the mean ± SD of four independent experiments ($n$ = 4). ns, not significant. **d**–**f** RT-qPCR analyses of encapsidated viral genome copy numbers in the media of iBCBL-1 cultures left untreated or treated with Dox for 2 days together with 10 μM TAT or TAT-PD1 peptides (**d**), 20 μM liensinine or vehicle (DMSO) (**e**), and Mdivi-1 (**f**) at the indicated concentrations. Mean ± SD, $n$ = 3. The $p$-value was determined by matched pair $t$-test (*$p$ < 0.05, **$p$ < 0.01, and ***$p$ < 0.001). Source data are provided as a Source Data file

clearance in HeLa.Kyoto cells. This cooperation is likely to be mediated by vIRF-1 activation of NIX-mediated mitophagy rather than vice versa in that vIRF-1 utilized the LC3-interacting motif of NIX for mitochondrial clearance (Fig. 5g).

How does vIRF-1 activate NIX-mediated mitophagy? NIX forms a unique dimeric structure resistant to ionic detergents (e.g., sodium dodecyl sulfate) and reducing reagents (e.g., dithiothreitol) and its dimerization is functionally implicated in apoptosis and possibly mitophagy[43–45]. However, the precise details of the mechanisms by which dimerization contributes to the functions of NIX remain elusive. In fact, when overexpressed, NIX was able to readily dimerize but did not induce mitochondrial clearance by itself (Supplementary Fig. 3 and Fig. 3g). Moreover, co-transfected vIRF-1 induced mitochondrial clearance without promoting further dimerization of NIX. However, the NIX variants NIX-TA$^{VAMP1B}$ and NIX-TA$^{FIS1}$, containing heterologous tail-anchor sequences, lost the ability to form a

homodimer and to induce mitochondrial clearance when co-expressed with vIRF-1, although they could localize to mitochondria and bind to vIRF-1 (Fig. 5). Based on these data, NIX dimerization may be a prerequisite but not sufficient to mediate vIRF-1-activated mitophagy. A recent report showed that the phosphorylation of serine residues adjacent to the LIR of NIX stabilized the interaction with LC3B and promoted NIX-mediated mitophagy[46]. As vIRF-1 did not enhance the interaction of NIX and LC3B (Fig. 5h), however, it is unlikely that vIRF-1 induces the NIX phosphorylation. Nevertheless, it is worth investigating in future studies if vIRF-1 promotes NIX interactions with ATG8 family members other than LC3B. It is also conceivable that vIRF-1 activates other mitophagic pathways mediated by NIX. For example, NIX serves as a substrate of the E3 ubiquitin ligase PARK2, and then ubiquitinated NIX recruits the autophagy receptor NBR1 to the mitochondria[47]. Intriguingly, NIX was also shown to induce lysosomal degradation of mitochondria via

MIEAP-induced accumulation of lysosome-like organelles within mitochondria, designated as MALM[48]. For future investigations, it would be interesting to examine if vIRF-1 also activates the non-canonical pathways of NIX for mitochondrial clearance.

In this study, we have also discovered a role of vIRF-1 in inducing mitochondrial fragmentation, potentially via a mitochondrial fission protein, DRP1, although the exact mechanism of action remains to be elucidated. It has been reported that mitochondrial fission requires translocation of DRP1 from the cytosol to mitochondria[49]. However, DRP1 was readily detected in the mitochondria of latent BCBL-1 cells and its level did not increase upon virus lytic replication (Fig. 3a). Recent studies have revealed that DRP1 is subject to multiple post-translational modifications (PTMs), including phosphorylation, acetylation, ubiquitination, SUMOylation, and O-linked N-acetyl-glucosamine glycosylation, to modulate its stability, localization, and activity[50]. Thus, it is worth investigating in the future if vIRF-1 and lytic replication modulate the PTMs of DRP1 to promote mitochondrial fragmentation in HHV-8-infected cells.

Asymmetric mitochondrial fission, in which damaged or dysfunctional mitochondrial components are partitioned from intact components, has been postulated to play a key role in processing of the altered mitochondrial components via mitophagy[51–53]. Interestingly, BNIP3 can induce both mitochondrial fragmentation and mitophagy[54,55]. Notably, DRP1 binding to BNIP3 and recruitment to mitochondria and mitochondrial fission appear to be a prerequisite for BNIP-mediated mitophagy in cardiomyocytes. Whereas NIX shares about 52% sequence identity (64% sequence similarity) with BNIP3, it is unclear whether NIX also plays a role in mitochondrial fission. Our data showed that overexpression of NIX often induced a clustering of mitochondria at the perinuclear area rather than fragmentation (Fig. 3g and Supplementary Fig. 6). Thus, it is likely that vIRF-1-induced mitochondrial fission may contribute concomitantly to the initiation of NIX-mediated mitophagy during virus replication.

It is currently unclear whether mitochondria are functionally altered and the damaged mitochondria trigger mitophagy during HHV-8 lytic replication. Here, we have demonstrated that an accumulation of mitochondria content is cytotoxic to lytic cells and leads to amplification of antiviral responses, including apoptosis and innate immune responses. Our findings suggest that mitochondria-localized vIRF-1 contributes to global regulation of the antiviral responses to HHV-8 lytic replication by activating NIX-mediated mitophagy, as well as specific regulation of the antiviral responses via its inhibitory interactions with proapoptotic BOPs and MAVS. Furthermore, this study identifies the vIRF-1-derived PD1 peptide as an anti-HHV-8 agent, potentially providing a basis for development of antiviral and therapeutic treatments.

## Methods

**Cell culture.** BCBL-1 (ATCC), TRExBCBL-1-RTA (a gift from Dr. Jae U. Jung, hereafter simply termed iBCBL-1), and iBCBL-1 derivative lines were cultured in RPMI 1640 medium (Quality Biological) supplemented with 15% fetal bovine serum (FBS), stable L-alanyl-glutamine (Glutamine XL), antibiotics, including streptomycin and penicillin at 37 ℃ and 5% CO₂. 293T, HeLa.Kyoto (a gift from Dr. Ron R. Kopito), and iSLK and iSLK-BAC16 (a gift from Dr. Jae U. Jung) cells were cultured in Dulbecco's Modified Eagle Medium (DMEM) supplemented with 10% FBS, Glutamine XL, and antibiotics. The cell lines were tested for mycoplasma contamination (R&D systems) and if necessary cultured in plasmocin™ treatment or prophylactic (InvivoGen). Transient transfection with plasmids was performed using GenJet version II (SignaGen Laboratories) following the manufacturer's instructions. For stable and doxycycline (Dox)-inducible expression of short hairpin RNAs (shRNAs), iBCBL-1 cells were lentivirally transduced with the indicated shRNAs (Supplementary Table 3) in the presence of 10 μg/ml polybrene overnight and stably transduced cells were selected by growing in the presence of 1 μg/ml puromycin or 400 μg/ml geneticin for >1 month, and pooled clones were collected. Transfection of iSLK cells with bacmids BAC16 and BAC16.ΔPD was performed using Fugene® HD (Promega) as described by the manufacturer's

protocol and the transfected cells were selected in the presence of hygromycin B (1200 μg/ml) together with geneticin (250 μg/ml) and puromycin (1 μg/ml). Reconstitution of iSLK/BAC16.ΔPD cells with vIRF-1 or empty vector was performed by lentiviral transduction in the presence of 10 μg/ml polybrene overnight.

**Lentivirus production.** To produce lentiviruses, 293T cells were co-transfected with the lentiviral transfer vector together with the packaging plasmid psPAX2 and the vesicular stomatitis virus G protein expression plasmid pVSV-G at a ratio of 5:4:1. Two days later, virions were collected from the culture medium by ultracentrifugation in an SW28 rotor at 25,000 rpm for 2 h at 4 ℃. The virion pellets were resuspended in an appropriate volume of phosphate-buffered saline (PBS) to achieve 100x concentration. The transduction unit of lentiviruses was determined in 293T cells in the presence of appropriate antibiotics.

**Isolation of mitochondria.** Pure mitochondria were isolated using Axis-Shield OptiPrep iodixanol (Sigma-Aldrich). Latent and lytic iBCBL-1 cells were homogenized in buffer B (0.25 M sucrose, 1 mM EDTA, 20 mM HEPES-NaOH [pH 7.4]) with 50 strokes of a Dounce glass homogenizer and centrifuged at 1000 × g for 10 min. An aliquot of homogenate was used as total-cell extracts. The supernatant was further centrifuged at 13,000 × g for 10 min. The pellet was used as a crude fraction enriched in mitochondria. For further enrichment, the pellet was resuspended in 36% iodixanol, bottom-loaded under 10% and 30% iodixanol gradients, and centrifuged at 50,000 × g for 4 h. The mitochondria were collected at the 10%/30% iodixanol interface. For isolation of mitochondrial detergent-resistant membrane microdomains (mDRM), crude or pure mitochondria were incubated in TNE buffer (50 mM Tris-HCl [pH 7.4], 150 mM NaCl, and 1 mM EDTA) containing 1% Triton X-100 on ice for 30 min, and centrifuged at 21,000 × g for 10 min. The supernatant was used as a soluble mitochondrial fraction, and the pellet was used as the mDRM fraction. The pellet was boiled in 1× sodium dodecyl sulfate (SDS) sample buffer.

**DNA manipulation.** All polymerase chain reaction (PCR) amplification and site-directed mutagenesis, including point and deletion mutations were performed using Platinum™ Pfx or SuperFi™ DNA polymerase (Thermo Fisher Scientific). Subcloning of open reading frames (ORFs) and their derivatives into expression plasmids including pICE (a gift from Steve Jackson; Addgene plasmid #46960), pGEX-4T-1 (GE Healthcare Life Sciences), and NanoBiT® system vectors (Promega) was performed using appropriate restriction enzyme sites (Supplementary Table 1). Overlap extension PCR[56] was carried out for replacement of the NIX tail-anchor (TA) region with the TA region of other tail-anchored proteins and the fused genes were cloned into vectors pBiT1.1-N (Promega) and pICE_V5[33]. Primers are listed in Supplementary Table 3.

**Mutagenesis of HHV-8 bacmid BAC16.** The BAC16 genome was edited using a two-step seamless Red recombination in the context of E. coli strain GS1783 (a kind gift from Greg Smith)[57]. Briefly, PCR amplification was performed to generate a linear DNA fragment containing a kanamycin resistance expression cassette, an I-SceI restriction enzyme site, and flanking sequences derived from HHV-8 genomic DNA, each of which includes a 40-bp copy of a duplication. The direct connection of the translational initiation codon with codon 76, which results in deletion of the proline-rich domain (PD, 2 to 75 residues) of vIRF-1, was placed in the middle of the duplication (Supplementary Table 3 and Supplementary Fig. 7a). This fragment was purified and then electroporated into GS1783 cells harboring BAC16 and transiently expressing gam, bet, and exo, which are expressed in a temperature-inducible manner from the lambda Red operon in GS1783 chromosome. The integrated KanR/I-SceI cassette was cleaved by I-SceI enzyme that was inducibly expressed by treatment with 1% L-arabinose, resulting in a transiently linearized BAC16. A second Red-mediated recombination between the duplicated sequences results in recirculation of the BAC DNA and seamless loss of the KanR/I-SceI cassette. Kanamycin-sensitive colonies were selected via replica plating. The BAC DNAs were purified using the NucleoBond BAC 100 kit (Clontech). The recombination area was amplified by PCR and the mutation was verified by DNA sequencing of the PCR amplicon (Supplementary Fig. 7a). Gross genome integrity and the deletion of the PD region were verified using digestion with AvrII and SpeI restriction enzymes and agarose gel analysis of digestion profiles (Supplementary Fig. 7b).

**Immunological assays.** Antibodies used in immunological assays including immunoblotting, immunoprecipitation, and immunostaining are listed in Supplementary Table 2. For the preparation of total-cell extracts, cells were resuspended in RIPA buffer (50 mM Tris [pH 7.4], 150 mM NaCl, 1% Igepal CA-630, and 0.25% deoxycholate) containing a protease inhibitor cocktail and protein phosphatase inhibitors, including 10 mM NaF and 5 mM Na₃VO₄ and sonicated using Bioruptor (Diagenode) for 5 min in ice water at a high-power setting (320 W). For immunoblotting, cell lysates were separated by sodium dodecyl sulfate polyacrylamide gel electrophoresis (SDS-PAGE), transferred to nitrocellulose or polyvinylidene difluoride membranes, and immunoblotted with appropriate primary antibodies diluted in SuperBlock™ (PBS) blocking buffer (Thermo Fisher Scientific). Following incubation with horse radish peroxidase-labeled appropriate

secondary antibody, immunoreactive bands were visualized by an enhanced che-miluminescent (ECL) reagent on an ECL film. ImageJ software (NIH) was used to quantify the signal from immunoblots. For immunoprecipitation (IP), total-cell or mitochondrial extracts were incubated with specific primary antibody at 4 ℃ overnight and incubated with protein G-agarose beads (Cell Signaling Technology) for an additional 3 h. Immunoprecipitants were washed with RIPA buffer, followed by elution of bound proteins with 1× SDS sample buffer. To avoid detection of IgG used in IP assays, Clean-Blot IP detection reagent (Thermo Fisher Scientific) was used. For indirect immunofluorescent assay (IFA), cells grown on a coverslip (and transfected) were fixed in Image-iT™ fixative solution (Thermo Fisher Scientific) and permeabilized in 0.5% Triton X-100 prepared in PBS. iBCBL-1 cells were attached on poly-L-lysine coated coverslips. Following incubation with SuperBlock™ PBS blocking buffer for 1 h at room temperature, coverslips were incubated with primary antibodies, washed with PBS, and then incubated with appropriate fluorescent dye-conjugated secondary antibodies. Coverslips were mounted in ProLong™ Gold Antifade Mountant containing 4′,6-diamidino-2-phenylindole (DAPI) (Thermo Fisher Scientific) on glass slides and cells were imaged on a Zeiss 700 confocal laser scanning microscope (LSM) with a 40x or 63x oil-immersion objective and Zen software. For IFA analysis of mitochondria content (TFAM), images containing both untransfected and transfected cells were taken randomly under the microscope, and fluorescence intensity of TFAM was measured using ImageJ.

**Quantification of mitochondrial DNA**. The amount of mitochondrial DNA (mtDNA) in lytic iBCBL-1 cells was determined using a combined approach of IFA and fluorescence in situ hybridization (FISH). Firstly, iBCBL-1 cells were treated with Dox for 2 days and IFA was performed with rabbit anti-vIRF-1 antibody as described above to identify lytically infected iBCBL-1 cells. After incubation with an anti-rabbit IgG Alexa 594-conjugated secondary antibody, the cells were post-fixed in Image-iT™ fixative solution for 15 min in the dark. After washing in PBS three times, the cells were incubated with 0.1 μg/μl RNase A for 1 h at 37 ℃, permeabilized in ice-cold 0.7% Triton X-100 and 0.1 M HCl for 10 min on ice, and denatured in 50% formamide and 2x SSC at 80 ℃ for at least 30 min. Coverslips were dehydrated by successive rinsing in 70%, 80%, and 95% ethanol and air-dried, and then hybridized with mtDNA-specific probes overnight at 37 ℃ in a dark and humid chamber. The probes were prepared by nick translation of six overlapping PCR products (see (Supplementary Table 3 for primer sequences) representing the entire human mtDNA using FISH Tag™ DNA green kit (Thermo Fisher Scientific). Following washing, coverslips were mounted as above, images containing both vIRF-1-positive and -negative cells were taken randomly under the microscope, and fluorescence intensity of mtDNA was measured using ImageJ.

**Image and flow cytometric analyses of mitochondria content**. For cytometric analyses of mitochondria content, cells were immunostained in suspension with antibodies to mitochondrial proteins, including MTCO2 and TOM20. Briefly, cells were fixed in 4% formaldehyde at 37 ℃ for 10 min and permeabilized in a final concentration of 90% methanol. The fixed cells were washed twice with blocking buffer (0.5% BSA in PBS) by centrifugation at 3000 x g for 5 min and incubated with the indicated primary antibody in blocking buffer for 1 h at room temperature. Cells were washed twice with blocking buffer and stained with Alexa Fluor® 488 or 647-conjugated goat anti-mouse IgG antibody in blocking buffer. Image and flow cytometric data of the immunostained cells were acquired using Cellometer Vision CBA Image Cytometer (Nexcelom) and FACSCalibur (BD Biosciences), respectively. FCS Express 6 Flow software (De Novo Software) was used for data analysis.

**Assessment of mitolysosome formation**. For assessment of mitolysosome for-mation in iBCBL-1 cells, mitochondria and lysosomes were labeled using Cell-Light® reagent BacMam 2.0 (Thermo Fisher Scientific) as described in the manufacturer's protocol. Briefly, iBCBL-1 cells were left untreated or treated with Dox and 24 h later infected with baculoviruses encoding CellLight® Lysosomes-GFP and CellLight® Mitochondria-RFP at a ratio of 30 particles per cell. Twenty-four hours later, cells were fixed and visualized on the Zeiss 700 confocal LSM. For assessment of mitolysosome formation in BAC16 iSLK cells, IFA was performed using antibodies to TOM20 and LAMP1 along with respective secondary anti-bodies conjugated to Alexa Fluor® 594 and Alexa Fluor® 647 as described above.

**Electron microscopy**. Latent and lytic iBCBL-1 cells were fixed in buffer con-taining 2.5% glutaraldehyde, 3 mM MgCl₂, 0.1 M sodium cacodylate [pH 7.2] for 1 h at room temperature. After rinsing, samples were post-fixed in buffer containing 1% osmium tetroxide, 0.8% potassium ferrocyanide, 0.1 M sodium cacodylate for 1 h on ice in the dark. After rinsing with 0.1 M maleate buffer [pH6.2] at 4 ℃, samples were stained in 2% uranyl acetate in 0.1 M maleate buffer at 4 ℃ for 1 h in the dark, dehydrated in a graded series of ethanol and embedded in Eponate 12 (Ted Pella) resin, and polymerized at 60 ℃ overnight. Thin sections, 60 to 90 nm, were cut with a diamond knife on the Reichert-Jung Ultracut E Ultramicrotome and picked up with 2 x 1 mm formvar coated copper slot grids. Grids were imaged on a Philips CM120 at 80 kV and captured with an AMT XR80 CCD (8 megapixel) camera.

**Proximity ligation assay**. Proximity ligation assay (PLA) was performed using the DuoLink® In Situ Red Starter Kit Mouse/Rabbit (Sigma-Aldrich) following the manufacturer's instruction. Briefly, HeLa.Kyoto cells were seeded at low density on glass coverslips and 24 h later transfected with vIRF-1 and/or NIX plasmids. The next day, the cells were fixed in Image-iT™ fixative solution and permeabilized in 0.5% Triton X-100 prepared in PBS for 15 min. Samples were incubated with 3% bovine serum albumin (BSA) for 1 h at 37 ℃ in a humidified chamber and then overnight at 4 ℃ with rabbit anti-vIRF-1 and mouse anti-NIX (H-8, Santa Cruz Biotechnology) antibodies. Slides were then incubated for 1 h at 37 ℃ with a mix of the minUS (anti-mouse) and PLUS (anti-rabbit) PLA probes. Hybridized probes were ligated using the Ligation-Ligase solution for 30 min at 37 ℃ and then amplified using Amplification-Polymerase solution for 100 min at 37 ℃. Slides were then mounted using DuoLink® II Mounting Medium with DAPI and imaged on the Zeiss 700 confocal LSM.

**GST-pull-down assays**. Recombinant glutathione-S-transferase (GST) and GST-fusion proteins were expressed in BL21 derivative Rosetta cells (Novagen) and purified by standard methods. One microgram GST or GST-fusion protein was incubated with 20 μl bed volume of washed glutathione sepharose-4B beads for 1 h at room temperature. After washing in binding buffer (PBS plus 1% Triton X-100), the protein-bead complexes were incubated with 1 μg recombinant vIRF-1 (for GST-NIX) as described previously[25] or 1 μg recombinant NIX (Sino Biological, 12488-HNCE, for GST-vIRF-1), or 293T cell extracts containing vIRF-1-Flag proteins (for GST-NIX) at 4 ℃ overnight, washed in binding buffer four times, separated on SDS-PAGE, and subjected to immunoblotting.

**Protein fragment complementation assay (PCA)**. PCA was performed using NanoBiT® Protein:Protein Interaction System (Promega). Twenty-four hours after transfection of 293T cells in a six-well plate with the indicated genes in the NanoBiT vectors, cells were washed in PBS and resuspended in 1 ml of Opti-MEM™ I reduced serum medium (Thermo Fisher Scientific), and 100 μl of the cell suspension was transferred to a 96-well white plate in triplicate. Furimazine (N1110, Promega), a NanoLuc®-luciferase substrate, was diluted in PBS at a ratio of 1 to 50 and 25 μl of the diluted substrate was added to each well. After 5 min of incubation, the luciferase activity in each well was measured by GloMax® 96 microplate luminometer (Promega).

**Apoptosis assays**. Dead cells existing prior to the experiments were removed using Histopaque-1077 (Sigma) or Dead Cell Removal kit (Miltenyl Biotec). Apoptotic cells were identified by staining with FITC-annexin-V as described by the manufacturer's protocol (BioLegend). Also, apoptotic cell death was monitored by detecting the cleaved product of PARP1 via immunoblotting. Terminal deox-ynucleotidyl transferase (TdT)-mediated dUTP nick end labeling (TUNEL) was performed using Cell Meter™ TUNEL Apoptosis Assay kit as described by the manufacturer's protocol (AAT Bioquest).

**Real-time-quantitative PCR (RT-qPCR) analysis of gene expression**. Total RNAs were isolated using the RNeasy mini kit (Qiagen). First-strand cDNA was synthesized from 1 μg of total RNA using SuperScript IV reverse transcriptase (Thermo Fisher Scientific) with random hexamers. RT-qPCR was performed using an ABI Prism 7500 system (Applied Biosystems) with the FastStart SYBR green/ROX master mix (Sigma-Aldrich). Primers are listed in Supplementary Table 3. Reactions were performed in a total volume of 25 μl and contained 50 ng of reverse-transcribed RNA (based on the initial RNA concentration) and gene-specific primers. PCR conditions included an initial incubation step of 2 min at 50 ℃ and an enzyme heat activation step of 10 min at 95 ℃, followed by 40 cycles of 15 s at 95 ℃ for denaturing and 1 min at 60 ℃ for annealing and extension.

**HHV-8 replication assays**. For determination of encapsidated HHV-8 genome copy number, viral DNA was purified using Quick-DNA™ Viral Kit (Zymo Research) following pretreatment of the culture supernatants containing HHV-8 virions with DNase I (New England BioLabs) at 37 ℃ overnight. RT-qPCR was performed as described above with LANA primers (Supplementary Table 3). BAC16 DNA was used as a standard for the calibration curve. Also, the expression levels of lytic antigens including ORF45 and K8.1 were determined by immunoblotting.

**Reagents**. Chemical reagents were purchased from the following companies: MG132, Cell Signaling Technology; doxycycline, bafilomycin A1, chloroquine, and sodium butyrate, MilliporeSigma; Liensinine, AK Scientific; Mdivi-1, Cayman Chemical. TAT and TAT-PD1 peptides were custom synthesized by Biomatik, Canada.

**Quantification and statistical analysis**. Statistical parameters including statistical analysis, statistical significance, and p-value are stated in the Figure legends and Supplemental Figure legends. Statistical analyses were performed using Prism 8 (GraphPad Software). Differences between controls and samples were determined

by matched pair $t$-test and were considered significant when the $p$-value was <0.05 ($p < 0.05$).

**Reporting summary**. Further information on research design is available in the Nature Research Reporting Summary linked to this article.

## Data availability

All relevant data are available from the authors upon reasonable request. The source data underlying Figs. 1a, 1b, 1d, 1e, 2a, 2c–e, 3a, 3c–h, 3g, 4a, 4e–h, 5c–h, 6e, 6g, 6k, 7c, 7f, 7i, 8a–f and Supplementary Figs. 1e, 2, 3, 4, 5b, and 5c are provided as a Source Data file.

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

## Acknowledgements

We thank Dr. Edward Harhaj for critical reading of this manuscript. This work was supported by National Cancer Institute (NCI) grant R01CA214131 to Y.B.C. and by National Institute of Allergy and Infectious Diseases (NIAID) grant R21Al117168 to Y.B. C. The Sidney Kimmel Cancer Center Flow Cytometry core facility, used for particular aspects of the study, is supported by Cancer Center Grant NCI CCSG P30 CA006973. This research was funded in part by a 2016-2017 developmental grant to Y.B.C. from the Johns Hopkins University Center for AIDS Research, funded by NIH program grant P30AI094189, and an NIH Shared Instrumentation grant (S10OD016374 to the JHU Microscope Facility).

## Author contributions

M.T.V. designed the study, performed experiments, and analyzed the results. B.J.S. performed experiments. J.N. provided valuable suggestions, discussed the results, and proofread the manuscript. Y.B.C. designed the study, performed the experiments, analyzed the results, and wrote the manuscript.

## Additional information

**Competing interests:** The authors declare no competing interests.

