## [Peer Review File · Nature Communications]

Reviewers' Comments:

Reviewer #1:

Remarks to the Author:

This article entitled, "Activation of NIX-mediated mitophagy and replication by an interferon regulatory factor homologue of human herpesvirus" by Vo et al examines the interaction between human herpesvirus 8 (HHV-8)-encoded viral interferon regulatory factor 1 (vIRF-1) and the mitophagy receptor NIX. The authors report that this interaction between vIRF-1 and NIX activates mitophagy and leads to enhanced viral production by suppressing cellular antiviral responses.

Overall, this study is very interesting and well-written, and many of the analyses are performed rigorously. Increasing reports are showing that viral manipulation of host autophagy, mitophagy, and mitochondrial dynamics are crucial in many different contexts of viral infection, and this article highlights a novel aspect that has the potential to be important for the field. However, the paper would be greatly improved if more quantifiable data were presented to support much of the microscopy images. Many elements of the data are only qualitative, yet much of the authors' claims rely on that information.

Major comments:

Fig 1a, b, c- It is unclear if loss of mitochondrial content in vIRF-1 expressing cells is actually due to mitophagy or if this is a byproduct of cells undergoing cytopathic effect, due to enhanced viral replication. Assays should be done to compare viability of -vIRF-1 vs +v-IRF-1 cells.

Fig 1g, h- Microscopy images throughout the paper should include some kind of quantification (such as in 2f), if no other follow-up quantifiable data is presented. This is important because much of the story relies on the notion that vIRF-1 and NIX interact to induce mitophagy/mito clearance, thus single images of a few representative cells is not sufficient.

Fig 2a- It is unusual that DRP1 is equal in both BCBL and iBCBL total and mito protein fractions. Generally, DRP1 is considered to be a mitochondrial fission protein, and rarely used as a loading control. I would imagine that mitochondrial DRP1 would be much higher in the Dox-treated iBCBL cells. The authors should discuss the implications of equal amounts of mito-localized DRP1 in these two conditions.

Fig 2f- Are these differences significant? All treatments in the -DOX group seem to reduce MTCO2 levels within the same range as the differences in the +DOX group. It is hard to determine if these differences are just noise.

Minor comments:

Many of the merged fluorescence microscopy images are difficult to interpret as they are. Individual channels should be shown alongside the merged images, so the reader can better resolve what is going on.

Signed,

Roberta A. Gottlieb

Reviewer #2:

Remarks to the Author:

In this manuscript, the author report that human herpesvirus 8 (HHV-8)-encoded viral interferon regulatory factor 1 (vIRF-1) activates NIX-mediated mitophagy and viral replication. First, they have observed that vIRF-1 promotes a decrease in mitochondrial content during lytic reactivation through induction of NIX-mediated mitophagy. vIRF-1 binds directly to mitochondria-associated NIX and they have identified the N-terminal PD region of vIRF1 responsible for NIX binding. In NIX, they have

determined the physical and functional interactions with vIRF-1. Furthermore, the authors have demonstrated that mitochondria-localized vIRF-1 promotes survival of lytic cells by promoting mitophagy and that a cell-penetrating vIRF-1 PD peptide impairs mitophagy and mitochondrial dynamics. Finally, they have shown that mitophagy is critical for HHV-8 productive replication.

The identification of a new function of vIRF-1 in promoting mitophagy via NIX to support HHV-8 replication is really interesting. The data are convincing, experiments were well designed and performed, and the conclusions raised by the authors are fully supported by the results.

I have however few comments:

- In figs 1b and 1c, only MTCO2 is shown to illustrate the decrease in mitochondrial content, other mitochondrial proteins (Tom20, cytochrome c...) should be shown.
- Is it possible to illustrate Fig 1e with WB?
- Lane 172, the authors wrote that they failed to detect the expression of BNIP3. Was it performed by WB or RT-qPCR? If it was by WB, have the authors checked the validity of this antibody? Since NIX and BNIP3 are paralog, BNIP3 may be also involved in the mitophagy induced by vIRF-1.
- In Fig 5b, mitochondrial content was assessed using TOM20 while in previous figs 1 and 2, MTCO2 was used. Why?
- In Fig 5e, BAC16.DPD cells showed stronger cytopathic effects (apoptosis) but in fig 5c, they do not really look like apoptotic cells. Likewise, in Fig 6c, cells treated with TAT-PD1 do not show any obvious signs of apoptosis

Reviewer #3:

Remarks to the Author:

In this study, the authors demonstrated that HHV-8-encoded vIRF-1 promotes mitochondrial clearance by activating mitophagy to support lytic replication of HHV-8. Mechanically, vIRF-1 directly binds to a mitophagy receptor, NIX, on the mitochondria and thus activates NIX-mediated mitophagy to promote mitochondrial clearance. Besides, this study identifies the vIRF-1-derived PD1 peptide as an anti-HHV-8 agent for treatment of HHV-8-associated diseases. Although it is quite interesting, some more important experiments need to be done to support their conclusions.

1. In Figs. 1a-c, the author desired to show that vIRF-1 down-regulates mitochondrial content during lytic reactivation. However, these data merely suggested that mitochondria content of cells were reduced following lytic reactivation of HHV-8.
2. The author represented that when vIRF-1 was depleted, lytic cells with higher levels of MTCO2 were increased in number in Fig. 1e, while vIRF-1 might not influence mitochondrial biogenesis in Fig. 1f. To further confirm this conclusion, the author also needs to provide the protein expression of MTCO2 and TFAM via WB.
3. In Fig. 2g, overexpression of both vIRF-1 and NIX, but not each individually, could induce a significant decrease in mitochondrial content, as determined by TFAM immunostaining. The author needs to quantify the amount of mitochondrial content.
4. In Figs. 3a and 3c, the author utilized co-immunoprecipitation and GST-pull down assays to show that vIRF-1 binds directly to mitochondria-associated NIX. These data all adopted NIX to co-precipitate or GST-pull down with vIRF-1. To further examine the conclusion, the author needs to verify that vIRF-1 can also co-precipitate and GST-pull down with NIX.
5. The author could also adopt IFA to further examine the interaction between vIRF-1 and NIX in Fig. 3.
6. In Fig. 5d, the author showed that the formation of mitolysosomes was evident in lytic BAC16 cells, but not in BAC16-negative and lytic BAC16. Δ PD cells. The author should examine whether the formation of mitolysosomes was evident in BAC16. Δ PD cells after reverting the expression of vIRF-

1.

7. The author utilized bright-field microscopy to assess cytopathic effects in BAC16 and BAC16.ΔPD cells in Fig. 5e. The author could utilize flow cytometry technology to quantify the number of apoptosis cells.

8. The author used TAT-only peptide as a control for TAT-PD1. However, the TAT-only peptide cannot be detected in Fig. 6b, suggesting that the TAT-only peptide is not a suitable control. The author should randomly fuse 1-25 amino acids to TAT-only peptide, which works as a better control.

9. In Fig. 7, the author demonstrated the importance of mitophagy promoted by vIRF-1 for productive replication of HHV-8. To further verify the conclusion, the author could also detect the mRNA expression of viral lytic genes.

Responses to points raised by the reviewers

➤ *We appreciate all reviewers' critical reading of and useful and constructive comments on our original manuscript. The manuscript has been carefully and rigorously revised according to the comments. Our detailed responses to the points are provided below (in italics). Newly added figures are highlighted in blue.*

Reviewer #1 (Remarks to the Author):

This article entitled, "Activation of NIX-mediated mitophagy and replication by an interferon regulatory factor homologue of human herpesvirus" by Vo et al examines the interaction between human herpesvirus 8 (HHV-8)-encoded viral interferon regulatory factor 1 (vIRF-1) and the mitophagy receptor NIX. The authors report that this interaction between vIRF-1 and NIX activates mitophagy and leads to enhanced viral production by suppressing cellular antiviral responses.

Overall, this study is very interesting and well-written, and many of the analyses are performed rigorously. Increasing reports are showing that viral manipulation of host autophagy, mitophagy, and mitochondrial dynamics are crucial in many different contexts of viral infection, and this article highlights a novel aspect that has the potential to be important for the field. However, the paper would be greatly improved if more quantifiable data were presented to support much of the microscopy images. Many elements of the data are only qualitative, yet much of the authors' claims rely on that information.

Major comments:

Fig 1a, b, c- It is unclear if loss of mitochondrial content in vIRF-1 expressing cells is actually due to mitophagy or if this is a byproduct of cells undergoing cytopathic effect, due to enhanced viral replication. Assays should be done to compare viability of -vIRF-1 vs +v-IRF-1 cells.

➤ *To address the reviewer's point, we first examined apoptosis in lytic vIRF-1-positive and vIRF-1-negative iBCBL-1 cells using TUNEL assay. We found that about 8% of lytically reactivated iBCBL-1 cells (Dox-treated for 2 days) were TUNEL-positive apoptotic cells, only about 3% of which were vIRF-1-positive (Fig. 1d), indicating that the decrease in mitochondria content in vIRF-1-positive cells is not due to apoptosis. Furthermore, the apoptosis inhibitor zVAD, which inhibited PARP cleavage induced by virus replication, could not inhibit lytic replication-induced decrease of mitochondria content (MTCO2) (Fig. 1e). Taken together, these additional results suggest that the loss of mitochondria content is not mediated by virus replication-induced apoptosis.*

Fig 1g, h- Microscopy images throughout the paper should include some kind of quantification (such as in 2f), if no other follow-up quantifiable data is presented. This is important because much of the story relies on the notion that vIRF-1 and NIX interact to induce mitophagy/mito clearance, thus single images of a few representative cells is not sufficient.

➤ *We agree with this comment. For quantification of mitochondria content in microscopy images, we measured the TFAM immunofluorescence intensities of transfected or untransfected single cells (n>20 each) from 4 to 5 randomly selected images using ImageJ software. Relative integrated intensity normalized to untransfected cells is presented as Box-Whisker plots (refer to*

Figs. 2g, 3h, 4g, 6c, and supplementary Fig. 4). We also added the percentage of mitolysosome-containing iBCBL-1 cells calculated from 8 to 10 randomly selected images (refer to Figs. 1j and 5f-g).

Fig 2a- It is unusual that DRP1 is equal in both BCBL and iBCBL total and mito protein fractions. Generally, DRP1 is considered to be a mitochondrial fission protein, and rarely used as a loading control. I would imagine that mitochondrial DRP1 would be much higher in the Dox-treated iBCBL cells. The authors should discuss the implications of equal amounts of mito-localized DRP1 in these two conditions.

➤ *DRP1 is known to translocate from the cytosol to the mitochondria in response to certain stimuli. However, our data showed that DRP1 protein was readily detected in the mitochondrial fractions derived from latent BCBL-1 cells, and its level did not increase even after lytic reactivation. Besides translocation to mitochondria, post-translational modifications (PTMs) of DRP1 are known to be important for DRP1 regulation of mitochondrial fission (refer to the review paper; Adaniya SM, et al. (2019) Am J Physiol Cell Physiol, 316:C583). Thus, we speculate that virus replication and/or vIRF-1 might regulate DRP1's PTMs to activate DRP1-mediated mitochondrial fission in HHV-8-infected cells even though this is beyond the scope of this study. We discuss this in detail in the Discussion section. In addition, we have removed the phase "loading control" and rephrased it in the text; "The levels of the mitochondrial fission protein DRP1 and the mitochondrial chaperone HSP60 remained unchanged (Fig. 2a)".*

Fig 2f- Are these differences significant? All treatments in the -DOX group seem to reduce MTCO2 levels within the same range as the differences in the +DOX group. It is hard to determine if these differences are just noise.

➤ *We have repeated the experiment using flow cytometry. Consistent with the previous data, vIRF-1-depleted BCBL-1 cells in the -Dox group still slightly reduced MTCO2 levels, but vIRF-1-depleted and even vIRF-1/NIX double depleted BCBL-1 cells in the +Dox group considerably increased MTCO2 levels compared to the previous result. We replaced the previous data with the new, much improved data (Fig. 2f).*

Minor comments:

Many of the merged fluorescence microscopy images are difficult to interpret as they are. Individual channels should be shown alongside the merged images, so the reader can better resolve what is going on.

➤ *We have added microscopy images with individual channels where necessary - Figs. 1a, 1d, 1j, 3b, 3c, 5d, and 6b.*

Reviewer #2 (Remarks to the Author):

In this manuscript, the author report that human herpesvirus 8 (HHV-8)-encoded viral interferon regulatory factor 1 (vIRF-1) activates NIX-mediated mitophagy and viral replication. First, they have observed that vIRF-1 promotes a decrease in mitochondrial content during lytic reactivation through induction of NIX-mediated mitophagy. vIRF-1 binds directly to

mitochondria-associated NIX and they have identified the N-terminal PD region of vIRF1 responsible for NIX binding. In NIX, they have determined the physical and functional interactions with vIRF-1. Furthermore, the authors have demonstrated that mitochondria-localized vIRF-1 promotes survival of lytic cells by promoting mitophagy and that a cell-penetrating vIRF-1 PD peptide impairs mitophagy and mitochondrial dynamics. Finally, they have shown that mitophagy is critical for HHV-8 productive replication.

The identification of a new function of vIRF-1 in promoting mitophagy via NIX to support HHV-8 replication is really interesting. The data are convincing, experiments were well designed and performed, and the conclusions raised by the authors are fully supported by the results.

I have however few comments:

- In figs 1b and 1c, only MTCO2 is shown to illustrate the decrease in mitochondrial content, other mitochondrial proteins (Tom20, cytochrome c...) should be shown.

➤ *We now include another mitochondrial protein, TOM20, in the mitochondria content assays (Figs. 1b and 1c). Also, we now include data showing a decrease in TFAM upon lytic replication in control but not in vIRF-1-depleted cells (Fig. 1i).*

- Is it possible to illustrate Fig 1e with WB?

➤ *We have added new data of MTCO2 immunoblotting to Fig. 1f and noted the relative band intensities of MTCO2 in lytic control and vIRF-1-depleted iBCBL-1 cells.*

- Lane 172, the authors wrote that they failed to detect the expression of BNIP3. Was it performed by WB or RT-qPCR? If it was by WB, have the authors checked the validity of this antibody? Since NIX and BNIP3 are paralog, BNIP3 may be also involved in the mitophagy induced by vIRF-1.

➤ *We had performed immunoblotting with BNIP3 antibody (Santa Cruz Biotechnology, sc-56167) to examine the expression of BNIP3 protein in BCBL-1 cells. According to the reviewer's suggestion, we validated the BNIP3 antibody with the lysate of 293T cells transfected with an expression vector for human BNIP3 (GenScript, OHu56510). Indeed, the BNIP3 antibody could readily detect transfected BNIP3, but not endogenous BNIP3 protein from 293T and both latent and lytic iBCBL-1 cells (Supplementary Fig. 2). In fact, we expected that BNIP3 may be detectable in the HHV-8-infected cells because hypoxia-inducible factor 1 alpha (HIF-1 α), a transcriptional factor involved in activation of BNIP3 expression, is known to be activated during HHV-8 infection (Shrestha P, et al. (2017) PLoS Pathogens, 13:e1006628). Nonetheless, we do not rule out the possibility that BNIP3 might be also involved in the mitophagy induced during virus replication and/or by vIRF-1.*

- In Fig 5b, mitochondrial content was assessed using TOM20 while in previous figs 1 and 2, MTCO2 was used. Why?

➤ *Previously, we had analyzed the levels of both TOM20 and MTCO2 in BAC16-infected iSLK cells using flow cytometry and obtained comparable results for them. Just as the histograms*

of TOM20 were in a narrow range rather than those of MTCO2, we decided to analyze TOM20 in subsequent experiments. Nonetheless, we included the MTCO2 data in Fig. 5a.

- In Fig 5e, BAC16.ΔPD cells showed stronger cytopathic effects (apoptosis) but in fig 5c, they do not really look like apoptotic cells. Likewise, in Fig 6c, cells treated with TAT-PD1 do not show any obvious signs of apoptosis

➤ *To examine the formation of mitolysosomes by IFA, iSLK and BAC16-infected iSLK cells were cultured on a cover slip in the presence or absence of lytic inducer. Apoptotic cells usually round up and detach from cover slips. Indeed, BAC16.ΔPD cells were more detached from cover slips after treatment with sodium butyrate and Dox compared to iSLK and BAC16 cells. For this reason, we could not observe apoptotic cells with fragmented nuclei (DAPI staining) from lytic BAC16.ΔPD culture. Nonetheless, we performed another apoptosis assay, Annexin V staining (Fig. 5i), in addition to PARP cleavage assay (Fig. 5j). The results confirmed that BAC16.ΔPD cells are more susceptible to lytic replication-induced apoptosis compared to iSLK and BAC16 cells.*

➤ *We previously showed that vIRF-1 PD1 peptide blocked mitochondrial clearance induced by overexpression of both vIRF-1 and NIX in HeLa.Kyoto cells, rather leading to a sequestration of mitochondria in the perinuclear areas (refer to Fig. 6c). However, we did not observe significant apoptosis in the HeLa cells co-transfected with vIRF-1 and NIX. This is consistent with the finding that the PD1 peptide induced an increase in mitochondrial content in latent BCBL-1 cells but did not induce evident apoptosis (refer to Figs. 6d(iii) and 6i). Furthermore, our new IFA experiments using anti-TAT antibody revealed that TAT-PD1, but not TAT, induced nuclear fragmentation (apoptosis) along with an accumulation of mitochondria in lytic BCBL-1 cells (Fig. 6b). Please note that, in the case of suspension culture such as iBCBL-1, all cells including normal and apoptotic cells were adhered to poly-L-lysine-coated cover slips after stimulation or reactivation. Taken together, these results suggest that a simple accumulation of mitochondria may not be sufficient to induce apoptosis without the addition of apoptotic inducers.*

Reviewer #3 (Remarks to the Author):

In this study, the authors demonstrated that HHV-8-encoded vIRF-1 promotes mitochondrial clearance by activating mitophagy to support lytic replication of HHV-8. Mechanically, vIRF-1 directly binds to a mitophagy receptor, NIX, on the mitochondria and thus activates NIX-mediated mitophagy to promote mitochondrial clearance. Besides, this study identifies the vIRF-1-derived PD1 peptide as an anti-HHV-8 agent for treatment of HHV-8-associated diseases. Although it is quite interesting, some more important experiments need to be done to support their conclusions.

1. In Figs. 1a-c, the author desired to show that vIRF-1 down-regulates mitochondrial content during lytic reactivation. However, these data merely suggested that mitochondria content of cells were reduced following lytic reactivation of HHV-8.

➤ *We agree with this comment. While vIRF-1 is the lytic protein of our interest and used as a marker of lytic cells in Fig. 1a, “vIRF-1-positive (or expressing)” does not necessarily mean that vIRF-1 regulates mitophagy. To help avoid this confusion, we removed a phrase “conceivably*

by vIRF-1” in the last sentence of the paragraph. Also, we changed the subtitle for the paragraphs of the result section to “Mitochondrial content is downregulated during lytic reactivation”.

2. The author represented that when vIRF-1 was depleted, lytic cells with higher levels of MTCO2 were increased in number in Fig. 1e, while vIRF-1 might not influence mitochondrial biogenesis in Fig. 1f. To further confirm this conclusion, the author also needs to provide the protein expression of MTCO2 and TFAM via WB.

➤ *We have performed immunoblotting analysis of MTCO2 and TFAM in control and vIRF-1-depleted iBCBL-1 cells. Consistent with cytometric data, immunoblotting showed that the levels of MTCO2 and TFAM protein levels increased in vIRF-1-depleted cells while they decreased in control (sh-Luc) cells during lytic replication (Figs. 1f and 1i).*

3. In Fig. 2g, overexpression of both vIRF-1 and NIX, but not each individually, could induce a significant decrease in mitochondrial content, as determined by TFAM immunostaining. The author needs to quantify the amount of mitochondrial content.

➤ *We have measured the TFAM immunofluorescence intensities of transfected or untransfected single cells (n>20 each) from 4 to 5 randomly selected images using ImageJ software. Relative integrated intensity normalized to untransfected cells is now presented as Box-Whisker plots (Figs. 2g, 3h, 4g, 6c, and supplementary Fig. 4).*

4. In Figs. 3a and 3c, the author utilized co-immunoprecipitation and GST-pull down assays to show that vIRF-1 binds directly to mitochondria-associated NIX. These data all adopted NIX to co-precipitate or GST-pull down with vIRF-1. To further examine the conclusion, the author needs to verify that vIRF-1 can also co-precipitate and GST-pull down with NIX.

➤ *We had previously performed the reciprocal co-immunoprecipitation (co-IP) experiments with the mitochondrial extract derived from lytic BCBL-1 cells. However, NIX protein was not readily detected in the vIRF-1 immunoprecipitated complex. We reasoned that our vIRF-1 antibody used in the co-IP would compete with NIX for vIRF-1 binding because the antibody was generated against the N-terminal region (1 to 22 amino acids) of vIRF-1, containing the residues (N8 and F10) required for NIX binding. Instead, we could detect vIRF-1 protein in NIX-immunoprecipitated complex as shown in Fig. 3a. In an effort to verify the intracellular interaction between vIRF-1 and NIX, we employed a proximity ligation assay (PLA, DuoLink) and a protein fragment complementation assay (PCA, NanoBiT) as shown in Figs. 3 and 4. Furthermore, we have added new data showing co-localization of vIRF-1 and NIX in lytic BCBL-1 and transfected cells (Figs. 3b and 3c).*

➤ *In accordance with the reviewer’s comment, we performed a reciprocal GST-pull down assay with purified recombinant NIX protein (commercially available from Sino Biological) and GST-vIRF-1 protein that we previously used (Choi YB and Nicholas J, (2010) PLoS Pathogens, 6:e1001031). The result has confirmed the direct interaction between vIRF-1 and NIX (Fig. 3f).*

5. The author could also adopt IFA to further examine the interaction between vIRF-1 and NIX in Fig. 3.

➤ *We have performed IFA assays in lytic BCBL-1 cells and transfected HeLa.Kyoto cells and could indeed detect co-localization of vIRF-1 and NIX in both systems (Figs. 3b and 3c). Additionally, we found that NIX protein appears to form aggregates in lytic vIRF-1-positive BCBL-1 cells (Fig. 3b).*

6. In Fig. 5d, the author showed that the formation of mitolysosomes was evident in lytic BAC16 cells, but not in BAC16-negative and lytic BAC16.ΔPD cells. The author should examine whether the formation of mitolysosomes was evident in BAC16.ΔPD cells after reverting the expression of vIRF-1.

➤ *We have performed IFA assays to examine mitolysosomes in lytic BAC16.ΔPD cells transduced with vIRF-1 and control empty vector. Indeed, vIRF-1 complementation promoted the formation of lytic reactivation-induced mitolysosomes along with inducing a sequestration of the mitochondria at the perinuclear area (Figs. 5f and 5g). In addition, we have examined lytic reactivation-induced PARP cleavage in empty vector and vIRF-1-transduced BAC16.ΔPD cells. The result showed, as expected, that transduced vIRF-1 protein was readily expressed and inhibited PARP cleavage induced by lytic replication (Fig. 5k). Taken together, these supplemental results further support our hypothesis that vIRF-1 inhibits lytic replication-induced apoptosis by activating mitophagy.*

7. The author utilized bright-field microscopy to assess cytopathic effects in BAC16 and BAC16.ΔPD cells in Fig. 5e. The author could utilize flow cytometry technology to quantify the number of apoptosis cells.

➤ *We have performed flow cytometry analysis of Annexin V staining in iSLK, BAC16, and BAC16.ΔPD cells that were left untreated or treated with lytic inducers for 2 days and the new data were included with quantification of the number of apoptotic cells (Fig. 5i). Consistent with the data from the PARP cleavage assay, a much higher percentage of apoptotic cells was detected in lytic BAC16.ΔPD cells (46%) compared to lytic BAC16 cells (18%).*

8. The author used TAT-only peptide as a control for TAT-PD1. However, the TAT-only peptide cannot be detected in Fig. 6b, suggesting that the TAT-only peptide is not a suitable control. The author should randomly fuse 1-25 amino acids to TAT-only peptide, which works as a better control.

➤ *In the original fig. 6b, we used anti-vIRF-1 antibody. In an effort to ascertain intracellular delivery of TAT-only peptide, we performed another immunoblotting with anti-HIV TAT (47-58) antibody (Santa Cruz Biotechnology, sc-65915). However, it was technically impossible for us to detect such a small molecule (~1 kDa) by immunoblotting. Instead, we performed IFA with the TAT antibody. Indeed, we could detect intracellularly delivered TAT-only peptide in lytic iBCBL-1 cells, particularly in the nuclei, while TAT-PD1 was detected specifically in the mitochondria (Fig. 6b). In fact, intracellular delivery of the TAT-only peptide has been demonstrated and the peptide has been used as a control by many other researchers. Nonetheless, in line with the reviewer's suggestion, a scrambled PD1 peptide (TAT-scPD1) was custom-synthesized: GRKKRRQRRRPOGGG-PGKCIIRAKGPOAMLGGOPPSDNFPP. Unfortunately, we found that TAT-scPD1 was not readily detected by immunoblotting with anti-TAT antibody, probably due to*

its instability within cell, while TAT-PD1 was readily detected in total cell extract and the mitochondrial detergent resistant microdomains (refer to the attached Fig. 1 below). As TAT-PD1 localized specifically to the mitochondria and inhibited vIRF-1 regulation of mitochondria content and fragmentation (Figs. 6c and 6g), we believe that TAT-PD1 could be a selective and competitive inhibitor of mitochondria-localized vIRF-1. Furthermore, our IFA showed that TAT-PD1 induced an aggregation and/or enlargement of the mitochondria along with enhanced nuclear fragmentation in lytic iBCBL-1 cells (see the inset of Fig. 6b). This is consistent with the electron microscopy data showing TAT-PD1-induced mitochondrial enlargement and nuclear fragmentation (Fig. 6e). In short, our studies using the TAT and TAT-PD1 peptides provide sufficient evidence to demonstrate the specific effects of PD1.

9. In Fig. 7, the author demonstrated the importance of mitophagy promoted by vIRF-1 for productive replication of HHV-8. To further verify the conclusion, the author could also detect the mRNA expression of viral lytic genes.

➤ We analyzed the transcript levels of viral lytic genes ORF45 and K8.1 using RT-qPCR. Unexpectedly, vIRF-1 and/or NIX depletions did not affect expression of these viral transcripts (Fig. 7c) even though they could significantly inhibit the respective protein levels and virus productive replication. As protein translation has been known to be rapidly down-regulated following induction of apoptosis compared to mRNA synthesis (Clemens MJ et al, (2000) Cell Death Differ 7:603; Lasfargues C et al, (2012) Int J Mol Sci 14:177), apoptosis induced by mitophagy inhibition in lytic cells appears to more readily affect the protein expression rather than the mRNA expression of lytic proteins.

Reviewers' Comments:

Reviewer #1:

Remarks to the Author:

This revised article by Vo et al is greatly improved with new data/experiments and addresses my concerns regarding the previous version. The data are now much more robust and convincing.

Reviewer #2:

Remarks to the Author:

The concerns raised by the different referees have been addressed in a really satisfactory manner. I therefore recommend acceptance for publication.

Reviewer #3:

None

REVIEWERS' COMMENTS:

Reviewer #1 (Remarks to the Author):

This revised article by Vo et al is greatly improved with new data/experiments and addresses my concerns regarding the previous version. The data are now much more robust and convincing.

➤ *Thank you.*

Reviewer #2 (Remarks to the Author):

The concerns raised by the different referees have been addressed in a really satisfactory manner. I therefore recommend acceptance for publication.

➤ *Thank you.*

Reviewer #3 (Remarks to the Author):

In this study, the authors demonstrated that HHV-8-encoded vIRF-1 promotes mitochondrial clearance by activating mitophagy to support lytic replication of HHV-8. Mechanically, vIRF-1 directly binds to a mitophagy receptor, NIX, on the mitochondria and thus activates NIX-mediated mitophagy to promote mitochondrial clearance. Besides, this study identifies the vIRF-1-derived PD1 peptide as an anti-HHV-8 agent for treatment of HHV-8-associated diseases. The manuscript is very dense, data are generally convincing in revised version and I believe the manuscript entitled “Activation of NIX-mediated mitophagy and replication by an interferon regulatory factor homologue of human herpesvirus” has met the requirements of Nature Communications for publication.

➤ *Thank you.*